# Multi-Objective Quantum-Inspired Seagull Optimization Algorithm

Yule Wang [1] , Wanliang Wang [1], Ijaz Ahmad [2,*] and Elsayed Tag-Eldin [3,*]

1. College of Computer Science and Technology, Zhejiang University of Technology, Hangzhou 310023, China; 1111712014@zjut.edu.cn (Y.W.); zjutwwl@zjut.edu.cn (W.W.)
2. Shenzhen College of Advanced Technology, University of Chinese Academy of Sciences (UCAS), Shenzhen 518055, China
3. Faculty of Engineering and Technology, Future University in Egypt, New Cairo 11835, Egypt
* Correspondence: ijaz@siat.ac.cn (I.A.); elsayed.tageldin@fue.edu.eg (E.T.-E.)

**Abstract:** Objective solutions of multi-objective optimization problems (MOPs) are required to balance convergence and distribution to the Pareto front. This paper proposes a multi-objective quantum-inspired seagull optimization algorithm (MOQSOA) to optimize the convergence and distribution of solutions in multi-objective optimization problems. The proposed algorithm adopts opposite-based learning, the migration and attacking behavior of seagulls, grid ranking, and the superposition principles of quantum computing. To obtain a better initialized population in the absence of a priori knowledge, an opposite-based learning mechanism is used for initialization. The proposed algorithm uses nonlinear migration and attacking operation, simulating the behavior of seagulls for exploration and exploitation. Moreover, the real-coded quantum representation of the current optimal solution and quantum rotation gate are adopted to update the seagull population. In addition, a grid mechanism including global grid ranking and grid density ranking provides a criterion for leader selection and archive control. The experimental results of the IGD and Spacing metrics performed on ZDT, DTLZ, and UF test suites demonstrate the superiority of MOQSOA over NSGA-II, MOEA/D, MOPSO, IMMOEA, RVEA, and LMEA for enhancing the distribution and convergence performance of MOPs.

**Keywords:** multi-objective optimization problem; Pareto front; quantum computing; seagull optimization algorithm; grid ranking





## 1. Introduction

Optimization is an indispensable and important application area in engineering applications and scientific research. Practical problems usually require the optimization of several objectives simultaneously, and these problems can be described as multi-objective optimization problems (MOPs). Unlike single-objective optimization problems, multi-objective optimization problems aim to find the optimal vectors in the decision space, known as the Pareto optimal solution (PS). Each objective of the PS cannot be better without the other objectives deteriorating. All the objective vectors of the PS form the Pareto optimal front (PF). Multi-objective optimization is one of the most difficult and popular problems in recent evolutionary computing research. Multi-objective optimization algorithms are designed to solve two key problems: (1) "How to search solutions whose objective vectors are on or near PF"; (2) "How to search solutions whose objective vectors are distributed as widely as possible along the PF."

Evolutionary computing has been widely used to solve MOPs with great success in many engineering problems. In recent years, many multi-objective optimization algorithms have been introduced by scholars. In particular, evolutionary computation and swarm intelligence algorithms have been introduced to solve MOPs through a process known as evolutionary multi-objective optimization, such as the fast and elitist multi-objective genetic

algorithm (NSGA-II) [1], the improving strength Pareto evolutionary algorithm for multi-objective optimization (SPEA2) [2], the multi-objective optimization evolutionary algorithm based on decomposition (MOEA/D) [3], and the non-dominated neighbor-based immune algorithm (NNIA) [4]. In addition, particle swarm optimization (PSO) is a type of swarm intelligence optimization algorithm. It has advantages such as fast convergence speed and strong robustness, and it has been successfully applied to solve single-objective optimization problems. Many scholars attempt to use the PSO algorithm to solve complicated and large-scale MOPs. Since the multi-objective particle swarm optimization algorithm (MOPSO) [5] was first proposed by Han, F.; Chen, many other improved Deep leaning model with MOPSO have been developed [6–8]. Cui et al. [9] proposed the multi-objective particle swarm optimization algorithm based on a two-archive mechanism (MOPSO_TA) to perform well in terms of convergence and diversity, simultaneously. Abdel-Basset et al. [10] improved and extended the whale optimization algorithm (WOA) to solve such multi-objective optimization problems (MIWOA). To solve increasingly complex multi-objective optimization problems in engineering practice, Wu et al. [11] proposed a multi-objective lion swarm optimization based on a multi-agent (MOMALSO). Zheng et al. [12] presented a dynamic multi-objective particle swarm optimization algorithm based on adversarial decomposition and neighborhood evolution (ADNEPSO) dealing with dynamic problems. Gu et al. [13] presented a random forest-assisted adaptive multi-objective particle swarm optimization (RFMOPSO) algorithm for expensive constrained combinatorial optimization problems.

Nowadays, multi-objective optimization algorithms are common for various real-world problems. Evolutionary computation and swarm intelligence algorithms have become important methods for optimization in the electronics field, such as for the optimal integration of electrical units in distribution networks [14], energy management strategies for range-extended electric vehicles [15], RFID network planning [16], multi_object fuse detection [17]. Multi-objective optimization algorithms also play a key role in cloud computing [18,19] supplier selection [20], and other combinatorial optimization problems [21].

Solving a multi-objective optimization problem generally leads to a set of Pareto non-dominated solutions. The optimization algorithm needs to find solutions as close as possible to the Pareto front while generating a solution set to cover the entire Pareto front as far as possible. Hence, multi-objective optimization algorithms need to balance the convergence of the algorithm with the distribution of Pareto optimal solutions. However, many multi-objective optimization algorithms are prone to local optimization, leading to unbalanced convergence and distribution problems. In order to counterpoise the convergence and distribution of Pareto optimal solutions, this paper proposes a multi-objective quantum-inspired seagull optimization algorithm (MOQSOA) to optimize the convergence and distribution of solutions in multi-objective optimization problems. This is particularly applicable in electronics fields such as: circuit design, electronics component arrangement, cost optimization, etc. MOQSOA is a hybrid algorithm combining a quantum-inspired search algorithm and the seagull optimization algorithm (SOA). Quantum-inspired search algorithms have adopted the principles and concepts of quantum mechanics including superposition, quantum gates, standing waves, and collapse, and are easy to find in local convergence in global searches. In addition, the SOA performs well in local searches, which helps maintain the balance between exploration and exploitation.

The contributions of this paper are summarized as follows. Firstly, an improved opposition-based learning strategy is used for the initialization of the seagull population to preserve distribution. Secondly, the current optimal solution is selected from the archive with global grid ranking and receives a real-coded quantum representation considered as a linear superposition of two probabilistic states, i.e., the positive and deceptive states. Thirdly, seagull individuals are updated with nonlinear migration, attacking operations, and quantum rotation gates for exploration and exploitation. In addition, the archive of non-dominated solutions is controlled with grid density ranking. The experimental results demonstrate the competitive performance of the proposed algorithm.

The remainder of the paper proceeds as follows: Section 2 surveys the theoretical background of multi-objective optimization problems, seagull optimization algorithm, and quantum computing. Section 3 details the proposed multi-objective quantum-inspired seagull optimization algorithm. Section 4 compares and discusses the performance of the proposed algorithm with several state-of-the-art metaheuristic algorithms. Finally, Section 5 draws the main conclusions and points out some possible future work.

## 2. Related Work

### 2.1. Multi-Objective Optimization Problems

Multi-objective optimization problems (MOPs), which involve more than one conflicting objective, can be described as follows [22]:

$$\min F(X) = (f_1(X), f_2(X), f_3(X) \cdots f_m(X))^T$$
$$\begin{cases} X = (x_1, x_2, \cdots, x_n)^{\mathrm{T}} \\ Y = F(X) \\ f_i : \boldsymbol{\Omega} \rightarrow \mathbb{R}^n \ (i = 1, 2, \cdots, m) \end{cases} \tag{1}$$

where the vector $x$ claims the decision space $X$, the objective function vector $F(X)$ includes $m$ ($m \geq 2$) objectives, $Y \subset \mathbb{R}^m$ represents the objective space, and $f : \mathbb{R}^n \rightarrow \mathbb{R}^m$ is the objective mapping function.

Pareto dominance: Given two vectors $x, y \in R^n$ and their corresponding objective vectors $F(x), F(y) \in \mathbb{R}^m$, $x$ dominates $y$ (denoted as $x \prec y$) if and only if $\forall i \in (1, 2, \ldots, m)$, $f_i(x) \leq f_i(y)$ and $\exists j \in (1, 2, \ldots, m)$, $f_i(x) < f_i(y)$.

Pareto optimal solution: A decision vector $x \in R^n$ is said to be Pareto optimal if and only if $\nexists y \in R^n : y \prec x$.

Pareto optimal set: The set of Pareto optimal solutions (PS) is called a Pareto optimal set if: $PS = \{x \in R^n | \nexists y \in R^n, y \prec x\}$.

Pareto optimal front: The Pareto optimal front (PF) is defined as: $PF = \{F(x) | x \in PS\}$.

### 2.2. Seagull Optimization Algorithm

The seagull optimization algorithm (SOA) [23] is a novel swarm optimization algorithm, proposed by Dhiman and Kumar in 2019, which simulates the migration and attacking behavior of seagulls. An extension of the SOA has been developed in terms of MOPs by introducing dynamic archive, grid, and leader-selection mechanisms [24,25].

Seagulls typically live in villages. They are able to locate and attack prey with their own knowledge. Migration and attacking actions are the most important actions of seagulls. They travel in groups during migration. Seagulls change their initial positions in order to prevent collisions. Seagulls will fly in a group in the direction of the fittest seagull with the best likelihood of survival. Other seagulls will update their initial positions based on the fittest seagull. Seagulls frequently attack migrating birds over the sea when they migrate from one place to another. They perform a spiral-shaped movement during attack.

The SOA mainly uses migration and attacking operations to simulate the migration and attacking behaviors of seagulls. The migration operation simulates how the group of seagulls move from one position to another with the exploration capability of the SOA. The attacking operation simulates how groups of seagulls hunt their prey with the exploitation capability of the algorithm.

### 2.3. Quantum Computing

Quantum computing is the combination of quantum mechanics in physics and computer science, and is an emerging theory of computing. Quantum-inspired evolutionary computing (QIEC) is a method based on concepts and principles from quantum mechanics. Narayanan and Moore [26] firstly combined evolutionary computation (EA) and quantum-inspired computation to solve traveling salesman problems (TSPs). After that, a series of EAs inspired by quantum computation appeared, such as the quantum-inspired evo-

lutionary algorithm (QEA) [27,28] and the quantum-inspired immune clonal algorithm (QICA) [29]. These kinds of algorithms are characterized by quantum bits and quantum gates. The quantum gates are used to change the quantum bits and generate new solutions through observing.

A quantum bit, known as qubit, is the smallest unit of information stored in quantum computing. A qubit may be in state "0" or state "1", or in a superposition of the two states. The state representation of a qubit in the Dirac notation can be given as:

$$|\psi\rangle = \alpha|0\rangle + \beta|1\rangle \tag{2}$$

where $\alpha$ and $\beta$ are complex numbers indicating the probability amplitudes of the respective states. $|\alpha|^2$ and $|\beta|^2$ denote the probability of observing a qubit in state "0" and state "1", respectively. The normalization of the states, resulting in unity, can be written as $|\alpha|^2 + |\beta|^2 = 1$.

Compared with classical bits, quantum bits can be in any superposition of two eigenstates of "0" and "1". Moreover, the superposition amplitudes of the two states can interfere with each other during quantum operation, which is called quantum interference. The principle of quantum superposition suggests that the system in a superposition of all of its possible states is described by probability density amplitudes. Additionally, all states can be processed in parallel to optimize the objective function.

A qubit individual consisting of a string of $m$ qubits can be described as follows:

$$\begin{bmatrix} \alpha_1 & \alpha_2 & \dots & \alpha_m \\ \beta_1 & \beta_2 & \dots & \beta_m \end{bmatrix} \tag{3}$$

where $|\alpha_i|^2 + |\beta_i|^2 = 1, i = 1, 2, \cdots, m$. Therefore, a quantum individual of length $m$ is capable of representing $2^m$ states simultaneously based on probability. Because a quantum individual can represent the superposition of several quantum bit states, a small population of quantum individuals can correspond to a large population of individuals under conventional representation.

Layeb [30] presented a new hybrid natural algorithm called the quantum-inspired harmony search algorithm (QIHSA) based on the harmony search algorithm (HSA) and quantum computing (QC). Another kind of quantum-inspired computation called the quantum particle swarm optimization algorithm (QPSO) was proposed by Sun et al. [31–34], inspired by the behavior of particles in a potential field. Particles are bounded by an attractor. Meanwhile, they appear anywhere in the space with different probability densities. Via setting potential well and solving the Schrödinger equation, a new style of search space is built. Based on this point, Li et al. [35] proposed an improved cooperative quantum-behaved particle swarm optimization method for solving real parameter optimization and obtained a good performance. QPSO was an improvement over the particle swarm algorithm based on the principles of quantum mechanics, which has better convergence properties than the ordinary particle swarm optimization algorithm [36].

In recent years, the combination of quantum computing and multi-objective problems has been studied and a number of new quantum multi-objective optimization algorithms have been proposed. Guo et al. [37] proposed a novel quantum-behaved particle swarm optimization algorithm with a flexible single-/multi-population strategy and a multi-stage perturbation strategy. At the first stage, the main target of the perturbation is to broaden the search range. The second stage applies the univariate perturbation to raise the local search accuracy. You et al. [38] presented a novel algorithm called DMO-QPSO, combining the quantum-behaved particle swarm optimization (QPSO) algorithm with the MOEA/D framework in order to make the QPSO able to solve MOPs effectively. Fan et al. [39] established a bi-level optimization model based on the quantum evolutionary algorithm and multi-objective programming to solve the problem of regional integrated energy systems. Hesar et al. [40] proposed a quantum-inspired multi-objective harmony search algorithm to solve multi-objective optimization problems. In this algorithm, a new

quantum mutation strategy is proposed, which is a combination of harmony improvisation operators and a quantum adaptive rotation gate. Dayana et al. [41] presented a Quantum Firefly Optimization-based Multi-Objective Secure Routing (QFO-MOSR) protocol for Fog-based WSN.

The QPSO algorithm has been applied in many real-life multi-objective problems due to the numerous variants of QPSO proposed. These methods have been successfully used to solve combinatorial optimization problems, such as scheduling problems [42,43], load forecast [44], routing optimization [45], disease diagnosis [46], and optimal design [47]. In addition, algorithms based on QPSO play important roles in other multi-objective problems, including multi-carrier communication [48] and system control [49].

When solving a multi-objective optimization problem, it is expected that the obtained solutions can fully reflect the entire Pareto front. However, it frequently occurs that solutions are only concentrated near a part of the Pareto front when solving practical problems. The key to resolving the issue is to deal with the balance of the convergence and distribution of Pareto optimal solutions. To address the above issue, this paper suggests a hybrid algorithm called the multi-objective quantum-inspired seagull optimization algorithm, termed MOQSOA, for multi-objective problems. In the MOQSOA, opposition-based learning is applied for initialization to preserve distribution. The current optimal solution is selected with global grid ranking, and receives a real-coded quantum representation considered as a linear superposition of positive and deceptive states. Additionally, individuals are updated with nonlinear migration, attacking operations, and quantum rotation gates.

## 3. The Multi-Objective Quantum-Inspired Seagull Optimization Algorithm

The main concept of the multi-objective seagull optimization algorithm (MOSOA) is based on the natural behavior of seagull populations. Four components have been used to develop the extension of the SOA in terms of MOPs: an archive controller, a grid mechanism, a leader-selection mechanism, and an evolutionary operator.

The evolutionary strategy explored in the seagull optimization algorithm is similar to most swarm intelligence algorithms. The deceptive nature of local optimal solutions, the loss of diversity, and weak causality present in the algorithm cause the algorithm to potentially fall into premature convergence. In this paper, a multi-objective quantum-inspired seagull optimization algorithm is presented for MOPs. The proposed algorithm combines opposite-based learning, the migration and attacking behavior of seagulls, grid ranking, and the superposition principles of quantum computing. The OBL mechanism is used to initialize the seagull population to obtain a better initialized population in the absence of a priori knowledge. To maintain a better balance between exploitation and the exploration of searching for global optimal solutions, the real-coded quantum representation of the current optimal solution and quantum rotation gate was adapted. Moreover, it contained the nonlinear migration and attacking operations of the SOA for exploration and exploitation. In addition, a grid mechanism with the global grid ranking (GGR) and the grid density ranking (GDR) provided a criterion for leader selection and archive control. The framework of the MOQSOA is shown in Figure 1, and the procedure of QDGWO and its main steps can be summarized as shown in Algorithm 1.

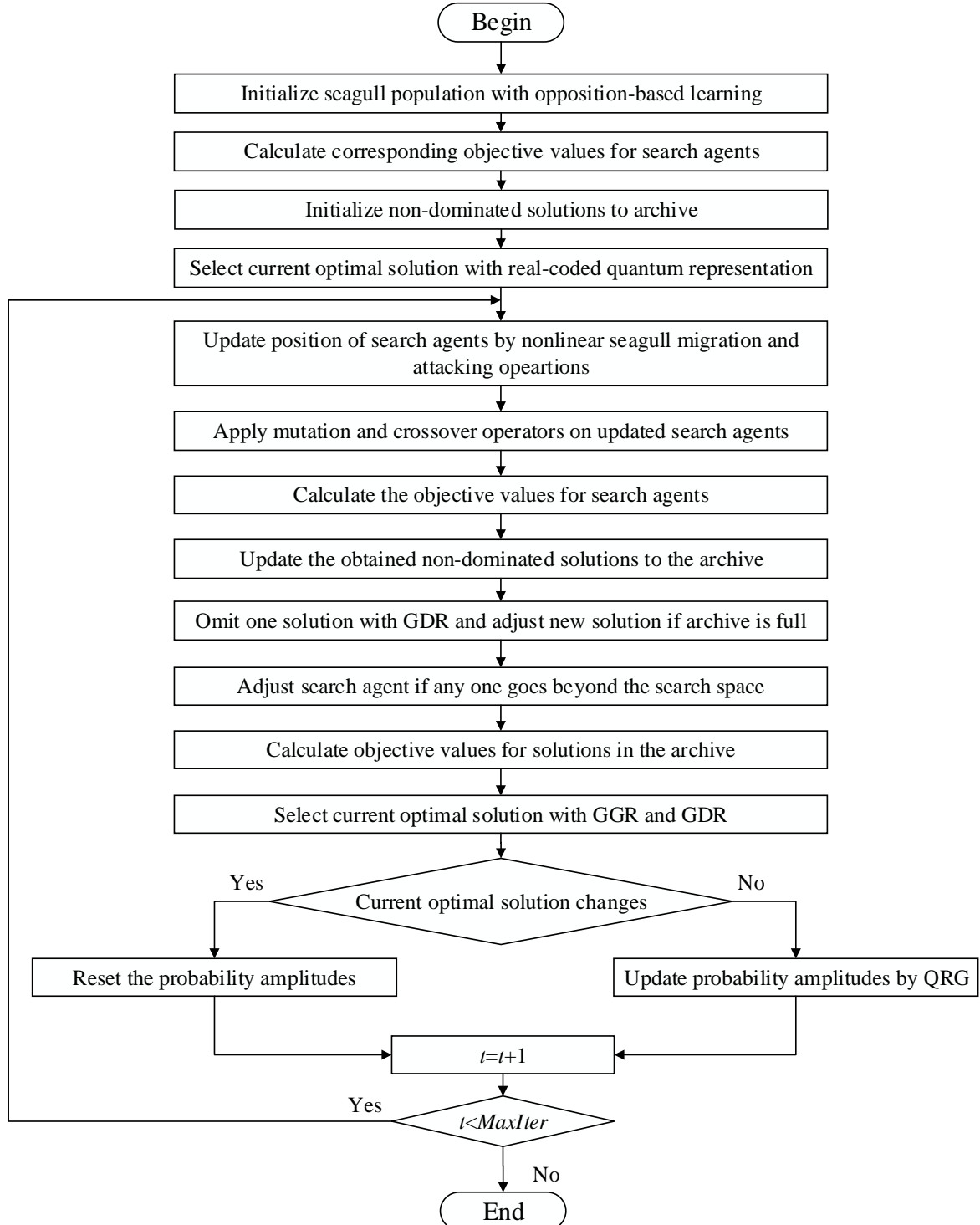

**Figure 1.** Framework of MOQSOA.

### 3.1. Initialization with Opposition-Based Learning

In the absence of a priori knowledge, initialization by random roulette in traditional evolutionary algorithms reduces the probability of sampling better regions in population-based algorithms. However, opposition-based learning (OBL) [50] can obtain more suitable initial candidate regions without a priori knowledge, thus increasing the probability of detecting better regions and promising potential to improve the fitness.

The opposite point of OBL can be defined as follows:

---

**Algorithm 1** Multi-Objective Quantum-Inspired Seagull Optimization Algorithm (MOQSOA)

---

**Input:** Seagull population $P$
**Output**: Archive non-dominated optimal solutions.
Initialize $P$ with opposition-based learning.
Calculate the corresponding objective values for each search agent.
Find all the non-dominated solutions and initialize these solutions to the archive of non-dominated optimal solutions.
Select the current optimal solution with global grid ranking and grid density ranking methods.
Encode the current optimal solution by real-coded quantum representation.
   **while** ($t < Maxiter$) **do**
      **for** each search agent **do**
         Update the position of current search agent by nonlinear seagull migration and attacking operations.
      **end for**
      Apply mutation and crossover operators on these updated search agents.
      Calculate the objective values for all search agents.
      Find the non-dominated solutions from the updated search agents.
      Update the obtained non-dominated solutions to the archive.
      **if** archive is full **then**
         Remove one of the most crowded solutions in the archive with the grid density ranking method.
         Add the new solution to the archive.
      **end if**
      Adjust search agent if any one goes beyond the search space.
      Calculate the objective values for each non-dominated solution in the archive.
      Select the current optimal solution with global grid ranking and grid density ranking methods.
      Conduct quantum update operation depending on whether the current optimal solution has changed or not.
      $t \leftarrow t + 1$
   **end while**
**return** archive of non-dominated optimal solutions
**end** MOQSOA

---

$P(x_1, x_2, \cdots, x_D)$ is given as a point in the $D$-dimensional space where $x_1, x_2, \cdots, x_D$ are real numbers and $x_i \in [a_i, b_i]$, $i = 1, 2, \cdots, D$. Then, its opposite point is defined as $\breve{P}(\breve{x}_1, \breve{x}_2, \cdots, \breve{x}_D)$ where

$$\breve{x}_i = a_i + b_i - x_i \tag{4}$$

MOQSOA firstly divides the population into two parts. Then, one part is generated by random initialization and the other part is generated by OBL. Later, the dominated solutions in the two parts are deleted, and the deleted individuals are replaced by random strategies. This specific process can be shown in Figure 2, and the main steps are shown as follows:

Step 1: The initial population $P_N$ is divided into two parts, named as $P_1$ and $P_2$. The individuals in the half population $P_1$ are randomly generated;

Step 2: The opposite points of individuals in $P_1$ are generated based on OBL and are added to $P_2$;

Step 3: After completing the construction of $P_2$, $P_1$ and $P_2$ are combined;

Step 4: The dominated solutions are deleted in $P_1 \cap P_2$;

Step 5: The deleted individuals are replaced by random strategies to generate the final initial population $P^*$.

The improved OBL strategy in the proposed algorithm is more suitable for multi-objective optimization problems. On the one hand, the improved OBL strategy can obtain a better initialized population because the dominated solutions between the original individuals and the opposite individuals have been removed. On the other hand, the distribution of the population can be guaranteed, because the original solution and its opposite solution are completely symmetric in the decision space. There must be a solution closer to the optimal solution between the original solution and its opposite solution, so the OBL strategy can improve the distribution of the initial population without a priori knowledge.

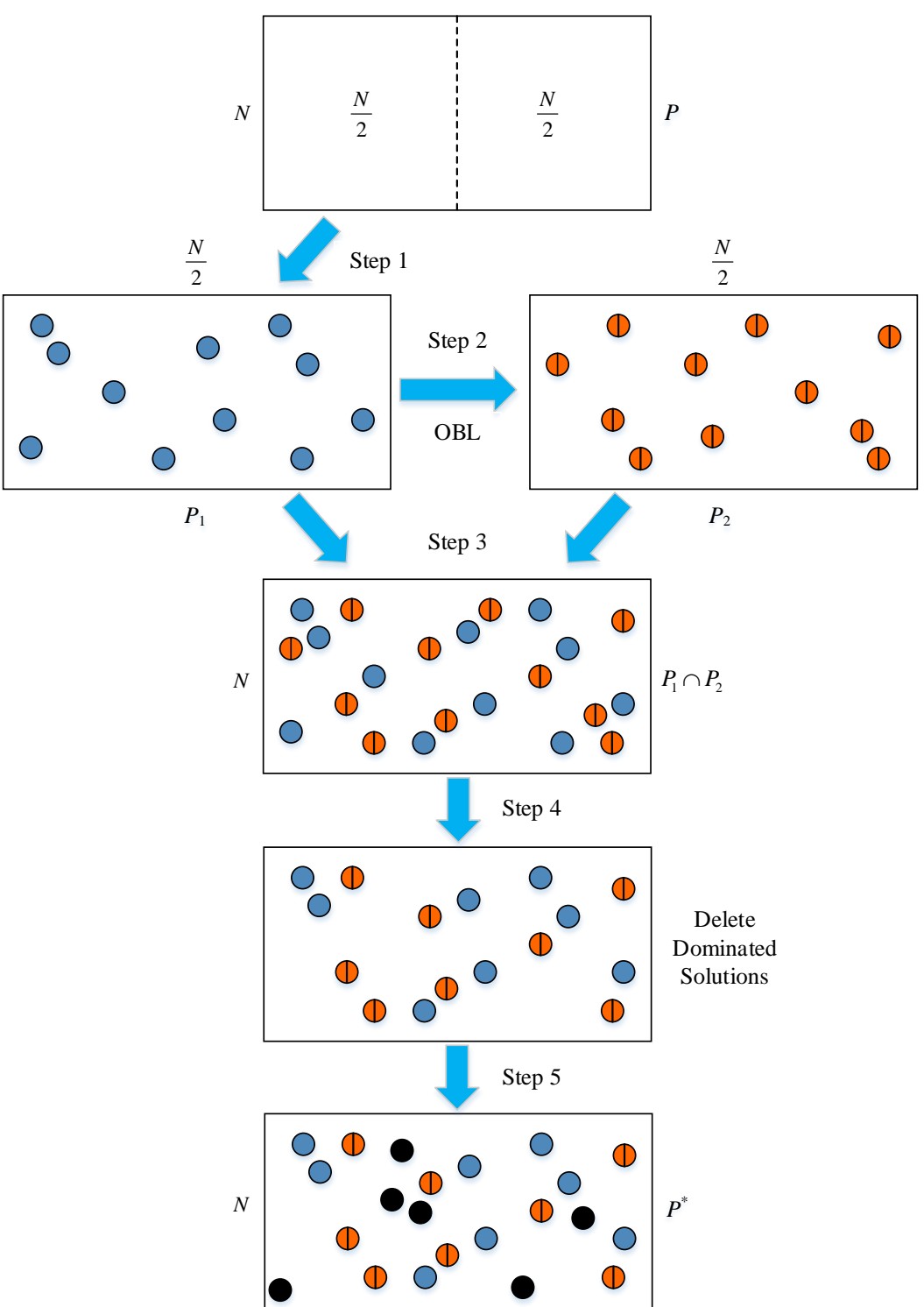

**Figure 2.** Initialization based on OBL.

## 3.2. Selection of Current Optimal Solution

The grid mechanism is a very efficient mechanism for characterizing and maintaining convergence and distribution. In addition, the grid mechanism can be utilized to show not only the superiority and inferiority between solutions, but also the differences in the objective values between optimal solutions and other solutions. In this paper, the global grid ranking (GGR) [22] is utilized to enhance the convergence of the algorithm, and the grid density ranking (GDR) is used to improve the distribution of the solutions.

The GGR represents the sum of the number of individual grid coordinates that are superior to other individuals in each objective. GGR is denoted as:

$$\text{GGR}(x_i) = \sum_{d=1}^{N_m} count(G_d(x_i) < G_d(x_j)) \tag{5}$$

where $x_i$ and $x_j$ are the two candidate solutions in the population which satisfy $i \neq j$; $G_d(x_i)$ is the grid coordinates of $x_i$ in the $d$th objective; $N_m$ denotes the number of problem objectives; and $count(\cdot)$ is the function to count the number to meet the conditions of $(\cdot)$. The larger the value of GGR($x_i$) is, the more individuals are dominated by $x_i$ in the sub-objectives.

The GDR is mainly used to view the crowding level around a candidate solution. A large GDR value indicates that the candidate solution is densely distributed with other solutions. GDR is generated by:

$$\text{GDR}(x_i) = count(\sum_{d=1}^{N_m} |G_d(x_i) - G_d(x_j)| < M) \tag{6}$$

where $x_i$ and $x_j$ are the two candidate solutions in the population which satisfy $i \neq j$; $G_d(x_i)$ is the grid coordinates of $x_i$ in the $d$th objective; $N_m$ denotes the number of problem objectives; $M$ is the number of current objectives; and $count(\cdot)$ is the function to count the number to meet the conditions of $(\cdot)$.

In a MOP, comparing new solutions with the existing solutions in a given search space is a key problem. The MOQSOA uses grid ranking to help compare the merits of solutions and select the best candidate solution. For candidate solutions in the archive, the algorithm prioritizes the candidate solution with a larger GGR value (i.e., dominating more solutions in the sub-objectives). Additionally, if the GGR values of several solutions are same, the solution with a smaller GDR value (i.e., lower crowding level) is preferred as the current optimal solution to guide the position updating of other individuals. If there are multiple solutions with the largest GGR values and the same GDR values, a roulette wheel is used to select a solution among these for the current optimal solution.

### 3.3. Real-Coded Quantum Representation of Current Optimal Solution

The deceptiveness of local optimal solutions is one of the main factors leading to premature convergence. The evolutionary strategy of the optimization algorithm tries to receive the gradient information by the direction of convergence. The reliability of the gradient information directly determines the effect of global convergence. A positive global optimal solution can accelerate the search process, while a deceptive local optimal solution prevents the exploration of the global optimum.

In this paper, the current optimal solution is considered as a linear superposition of two probabilistic states, i.e., the positive and deceptive states inspired by QEA. In the evolutionary process, every seagull individual makes its own judgment on the question of whether to accept the current optimal solution as the global optimum. If the $i$th seagull believes that the current optimal solution is a positive global optimal solution, then it will take the current optimal solution as the direction of convergence. Otherwise, it rejects and randomly chooses another individual as the direction of convergence.

At the beginning, it is assumed that the probability that the current optimal solution is positive or deceptive is equal. After several iterations, the positive probability of current optimal solution is enhanced if no changes have occurred, whereas if the current optimal solution is updated, the probability needs to be reset. The MOQSOA consists of two quantum operations, namely, the real-coded quantum representation of the current optimal solution and the quantum rotation gate for updating the probability amplitudes of two states.

The real-coded quantum representation [51] of an individual has been developed through the study of QEA [27]. In a binary-coded QEA, the qubit is used to represent

a linear superposition of state "0" and state "1" in a probabilistic manner. Similarly, a real continuous number is assumed to be in a deterministic state or a random state. In this paper, qubits are used to represent the global optimal solution and wave functions to calculate specific values.

A qubit can be represented by a state "0" (denoted as $|0\rangle$), or a state "1" (denoted as $|1\rangle$), or a linear superposition of both. The states of a qubit can be given by:

$$|\psi\rangle = \alpha|0\rangle + \beta|1\rangle \tag{7}$$

where $\alpha$ and $\beta$ represent the probability magnitudes of the two states, respectively, and satisfy $\alpha^2 + \beta^2 = 1$. $|\alpha|^2$ is the probability that the quantum bit is observed in state "0", and $|\beta|^2$ is the probability that the quantum bit is observed in state "1".

In quantum theory, a quantum state can be completely described by a wave function $w(x, t)$, which is a composite function of coordinates and time. Additionally, $|w(x, t)|^2$ is called the probability density, which implies the probability of the quantum state occurring at the appropriate location and time. Therefore, a normal wave function is introduced to calculate the observed values of real-coded quantum representation as:

$$|w(x_i)|^2 = \frac{1}{\sqrt{2\pi}\sigma_i} \exp\left[-\frac{(x_i - \mu_i)^2}{2\sigma_i^2}\right], i = 1, 2, \ldots, n \tag{8}$$

where $\mu_i$ is the expectation and $\sigma_i$ is the standard deviation. Here, Equation (8) is used to generate the position of a quantum individual after quantum observation, $\mu_i$ is the mean position of the individual, and $\sigma_i$ expresses the distribution range of the probability cloud around the mean position.

In this paper, the probability amplitude of the positive for the current optimal solution $P^t_{gb}$ is defined as $\alpha$, while the probability amplitude of the deceptive is defined as $\beta$. Since the two probability amplitudes satisfy $\alpha^2 + \beta^2 = 1$, the real-coded quantum representation of the current optimal solution can be expressed as:

$$P^t_{gb} \triangleq \begin{bmatrix} x_{gb,1} & x_{gb,2} & \cdots & x_{gb,n} \\ \alpha_1 & \alpha_2 & \cdots & \alpha_n \end{bmatrix} \tag{9}$$

where $n$ is the problem dimension, $\alpha_i$ is the probability amplitude by which the optimal solution component is considered positive, and $t$ is the current iteration.

### 3.4. Nonlinear Seagull Migration Operation

During migration, seagulls will move from their initial positions to the next positions within the group. The migration operation simulates this position movement process of the seagull population during exploration. The main concept of the MOQSOA is based on the SOA migration and attacking behaviors. Therefore, in the exploration phase, the seagull position movement process satisfies the following three steps: avoiding collisions, approaching the optimal neighbor's direction, and moving to the optimal search agent.

In order to avoid collisions with surrounding seagulls, an additional variable $A$ is employed to adjust the seagull's position:

$$C_s = A \times P_s(t) \tag{10}$$

where $C_s$ represents the direction of the search agent for no collisions with other search agents, $P_s$ represents the current location of the search agent, $t$ is the current iteration, and $A$ indicates the migration behavior of the seagull in the search space. In the basic seagull optimization algorithm, the size of $A$ is linearly decreased from parameter $f_c$ to 0 in the iteration:

$$A = f_c - \frac{f_c \cdot t}{t_{\max}} \tag{11}$$

where the value of $f_c$ is set to 2 in the basic seagull optimization algorithm [23].

However, in the actual optimization process, the search process shows a nonlinear curve downward trend. Therefore, if the control variable $A$ simulates the migration process of the seagull population in a purely linear decreasing manner, the actual search ability of the algorithm is affected. Therefore, this paper adopts a nonlinearly varying control variable $A$, which is more appropriate to the migration process of the actual seagull population:

$$A = f_c(2^\omega - 1) \tag{12}$$

$$\omega = e^{1 - \frac{t_{\max}}{t_{\max} - t}} \tag{13}$$

where the value of $f_c$ is set to 2, $t$ is the current iteration, and $t_{\max}$ is the maximum number of iterations.

The nonlinear variable $A$ can accelerate the convergence ability of the algorithm by rapidly decreasing in the early stage, and can improve the search accuracy of the algorithm by slowly decreasing in the later stage.

After ensuring that no collisions occur between seagulls, the seagull agents approach the best seagull. Here, each seagull individual makes an independent judgment on whether to recognize the current optimal solution as the global optimal solution (positive) or not (deceptive). If the seagull believes that the current optimal solution is a positive global optimal solution, it will take the position of the current optimal solution as the direction of convergence. Otherwise, it randomly chooses the direction of convergence.

Since a real-coded quantum representation is used to express the current optimal solution, the convergence direction of each agent is generated by:

$$M_s = B \times \left( \hat{P}_{gb}(t) - P_s(t) \right) \tag{14}$$

where $M_s$ denotes the convergence direction of individuals toward the best seagull, and $B$ is varied as:

$$B = 2 \times A^2 \times rand(0,1) \tag{15}$$

$\hat{P}_{gb} = \begin{bmatrix} \hat{x}_{gb,1} & \hat{x}_{gb,2} & \cdots & \hat{x}_{gb,n} \end{bmatrix}$ is the observed position of the current optimal solution. It is calculated as:

$$\hat{x}_{gb,i} = r_n \left[ x_{gb,i}, \sigma_i^2(|\psi_i\rangle) \right] (x_{i,\max} - x_{i,\min}) \tag{16}$$

where $r_n \left[ x_{gb,i}, \sigma_i^2(|\psi_i\rangle) \right]$ denotes a random number generated according to the wave function Equation (8), $x_{gb,i}$ is the expectation, and $\sigma_i^2(|\psi_i\rangle)$ is defined as:

$$\sigma_i^2(|\psi_i\rangle) = \begin{cases} 1 - |\alpha_i|^2, & \text{if } |\psi_i\rangle = |0\rangle \\ |\alpha_i|^2, & \text{if } |\psi_i\rangle = |1\rangle \end{cases} \tag{17}$$

where $\alpha_i$ is the probability amplitude that the optimal solution component is considered to be positive. The observation of $|\psi_i\rangle$ adheres to the following stochastic process:

$$|\psi_i\rangle = \begin{cases} |0\rangle, & \text{if } r_u \le |\alpha_i|^2 \\ |1\rangle, & \text{if } r_u > |\alpha_i|^2 \end{cases} \tag{18}$$

where $r_u$ is a uniform random variable between 0 and 1.

The schematic diagram of a seagull agent approaching the current optimal individual is shown in Figure 3. If the seagull individual recognizes the current optimal solution as the global optimal solution (positive identification), the observed position generated according to the wave function of Equation (8) will be in the vicinity of the current optimal individual, while if the seagull individual doubts the current optimal solution (deceptive identification), the individual randomly chooses its own search direction.

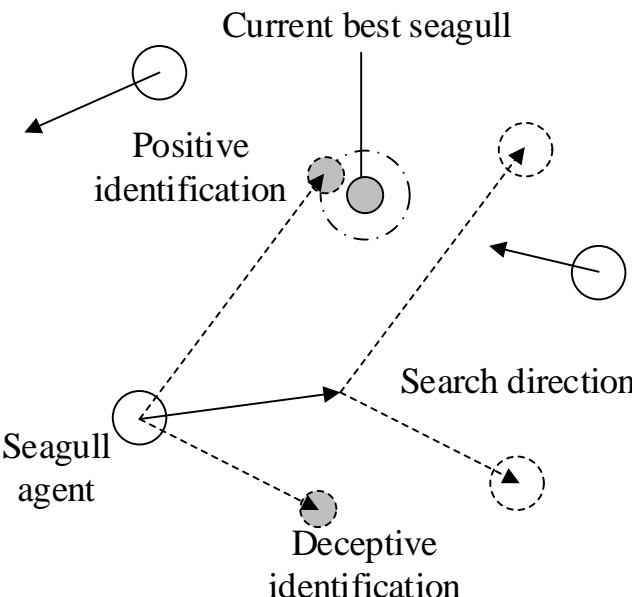

**Figure 3.** Schematic diagram of a seagull agent approaching the current optimal individual.

Finally, after calculating the direction of convergence for each agent, the seagulls update the position toward this direction:

$$D_s = |C_s + M_s| \tag{19}$$

where $D_s$ is the direction of migration for the seagulls composited by the direction for no collision ($C_s$) and the direction of the movement toward the best seagull ($M_s$).

### 3.5. Seagull Attacking Operation

Seagulls frequently attack other birds over the sea when migrating. They can maintain altitude during migration and constantly change their angle of attack and speed in flight. When it is necessary to launch an attack, the seagulls descend in a spiral through a three-dimensional space and move through the air by constantly changing their angle and radius. The attacking operation simulates the attacking process of the seagull population for exploitation.

The motion of the seagulls in the three-dimensional space is described as follows:

$$\begin{aligned} x &= r \times \cos(k) \\ y &= r \times \sin(k) \\ z &= r \times k \\ r &= u \times e^{kv} \end{aligned} \tag{20}$$

where $k$ is a random number in the range $[0, 2\pi]$, and $r$ is the spiral radius controlled by $u$ and $v$, which are usually taken as 1.

Combining the migration and attacking operations of the seagull, the overall seagull position is updated by:

$$P_s(t) = (D_s \times x \times y \times z) + \hat{P}_{gb}(t) \tag{21}$$

To obtain a better exploration and exploitation capability, the mutation and crossover operators, which are the same as those in NSGA-II [1], are employed in the MOQSOA.

### 3.6. Quantum Update Operation

The main update strategy in the QEA is the quantum rotation gate (QRG) [13]. The QRG is adopted as a quantum operator to update the probability amplitudes toward

the direction with better fitness. The probability amplitudes updated from QRG are calculated by:

$$\alpha_i(t+1) = \begin{bmatrix} \cos(\Delta\theta) & -\sin(\Delta\theta) \end{bmatrix} \begin{bmatrix} \alpha_i(t) \\ \sqrt{1 - [\alpha_i(t)]^2} \end{bmatrix} \tag{22}$$

where $\Delta\theta$ is the rotation angle, which is equivalent to the step size that determines the rate of convergence toward the current best solution.

Unlike the traditional update strategy of QEA, the QRG in this paper is an operator that enhances the probability amplitude of the positivity of the current optimal solution. For a given current optimal solution $P_{gb}^t$, the probability amplitude of positivity and deceptiveness will be initialized to $\alpha_i = \beta_i = \sqrt{2}/2$, $i = 1, 2, \ldots, n$. After iteration, if the current optimal solution remains optimal, the probability magnitude of positivity $\alpha_i$ will be increased by QRG, which means that the current optimal solution $P_{gb}^t$ will be more likely to be considered as the global optimal solution. Otherwise, the probability amplitudes will be re-initialized to remain vigilant to the deceptiveness of the local optimal solution.

To prevent the premature convergence of the quantum bits from falling into $|0\rangle$ or $|1\rangle$ (cannot escape the state by itself), a constant $\varepsilon$ close to 0 is applied for correction in this paper. The specific correction can be given by:

$$\alpha_i(t+1) = \begin{cases} \sqrt{\varepsilon}, & \text{if } \alpha_i(t+1) < \sqrt{\varepsilon} \\ \alpha_i(t+1), & \text{if } \sqrt{\varepsilon} < \alpha_i(t+1) < \sqrt{1-\varepsilon} \\ \sqrt{1-\varepsilon}, & \text{if } \alpha_i(t+1) > \sqrt{1-\varepsilon} \end{cases} \tag{23}$$

### 3.7. Archive Controller

All obtained Pareto optimal solutions are saved in a storage space called the archive. The archive controller decides whether to include a particular solution in the list or not. The algorithm compares the obtained objective values of the new solution with the individuals in the archive. The archive is updated with the following rules.

- If the archive is empty, the current solution will be accepted;
- If the new solution is dominated by an individual in the archive, then this solution should be discarded;
- If solutions in the archive are dominated by the new solution, then they are discarded from the archive. Additionally, the new solution will be accepted;
- If the new solution is not dominated by external solutions in the archive, then the particular solution should be accepted and stored within the archive. If the archive is full, then the solution with the largest GDR value is removed and the new solution goes into the archive for storage.

### 3.8. Algorithm Complexity

The MOQSOA employs strategies such as the seagull operator and real-coded quantum representation for finding the optimal solutions, and the computational complexity of the algorithm mainly comes from the archive controller of the non-dominated solutions. During the iteration, the complexity of comparison between the non-dominated solutions in the archive is $O(mN^2)$. Additionally, the complexity of the grid ranking mechanism is $O(mN^2)$, where $m$ is the number of objectives and $N$ is the number of population size. Therefore, the complexity of the MOQSOA is $O(mN^2)$. The complexity of the MOQSOA is equivalent to the NSGA-II, MOPSO, SPEA2 and other multi-objective algorithms.

### 4. Experimental Results and Discussion

In this section, the performance metrics and benchmark test functions sets used in the experiments are described. Then, the proposed MOQSOA is compared with three well-known and three state-of-the-art MOEAs named NSGA-II [1], MOEA/D [3], MOPSO [5], IMMOEA [52], RVEA [53], and LMEA [54] in order to evaluate the performance.

### 4.1. Experimental Setting

To evaluate the performance of the proposed algorithm, IGD [55] and Spacing [55] metrics were selected for the quantitative assessment of the performance of the optimization algorithms. The IGD metric measures the average distance from the point in the Pareto front to the nearest solution in the approximate front obtained by the algorithm to access the convergence and distribution of the solutions. The smaller the IGD value is, the better the convergence and distribution of the solutions obtained by the algorithm are. The Spacing metric measures the range variance of the neighbor solutions in the non-dominated solutions by comparison with the solutions converged to the true Pareto front. The smaller the SP value is, the better the distribution of the solutions obtained by the algorithm.

To evaluate the efficiency of the proposed MOQSOA, the proposed algorithm was validated with standard benchmark test problems including ZDT [56], DTLZ [57], and UF [58]. The characteristics of these test problems are shown in Table 1.

**Table 1.** Characteristics of benchmark test problems.

| Test Problems | Properties | Number of Objectives |
|---------------|------------|----------------------|
| ZDT1 | Convex | 2 |
| ZDT2 | Concave | 2 |
| ZDT3 | Disconnected | 2 |
| ZDT4 | Convex | 2 |
| ZDT6 | Concave | 2 |
| DTLZ1 | Linear | 3 |
| DTLZ2 | Concave | 3 |
| DTLZ3 | Concave | 3 |
| DTLZ4 | Concave | 3 |
| DTLZ5 | Concave | 3 |
| DTLZ6 | Concave | 3 |
| DTLZ7 | Disconnected | 3 |
| DTLZ8 | Linear | 3 |
| DTLZ9 | Concave | 3 |
| UF1 | Convex | 2 |
| UF2 | Convex | 2 |
| UF3 | Convex | 2 |
| UF4 | Concave | 2 |
| UF5 | Disconnected | 2 |
| UF6 | Disconnected | 2 |
| UF7 | Linear | 2 |
| UF8 | Concave | 3 |
| UF9 | Disconnected | 3 |
| UF10 | Concave | 3 |

All experiments were conducted with Matlab R2016b and PlatEMO v2.9 [59] running on an Intel Core i7-4790 CPU @ 3.60 GHz and Windows 7 Ultimate Edition.

In the experiment, the size of the population and archive were set to 100. The maximum number of iterations in all cases was set to 1000. The parameters of the algorithms used in the experiments are presented in Table 2.

Thirty independent runs were executed for each test problem to avoid randomness. Moreover, the Wilcoxon signed-rank test [60] was adopted to compare the results obtained by the MOQSOA and the six compared algorithms in Tables 3 and 4. The test used a significance level $\alpha = 0.05$, and "+", "−", and "=" indicate that the algorithm is superior, inferior, or equal to the MOQSOA, respectively.

**Table 2.** Parameters of the algorithms in the experiment.

| Algorithm | Parameter | Value |
|---|---|---|
| NSGA-II | Crossover probability $p_c$ | 0.8 |
| | Mutation probability $p_m$ | 0.1 |
| MOEA/D | Number of neighbors $T$ | 10 |
| | Probability of selecting parents $p_p$ | 0.9 |
| | Distribution index $D_i$ | 30 |
| | Differential weight | 0.5 |
| MOPSO | Number of grids $nGrid$ | 10 |
| | Inertia weight $w$ | 0.5 |
| | Personal coefficient $c_1$ | 1 |
| | Social coefficient $c_2$ | 2 |
| IMMOEA | $K$ | 10 |
| RVEA | $\alpha$ | 2 |
| | $f_r$ | 0.1 |
| LMEA | $nSel$ | 5 |
| | $nPer$ | 50 |
| | $nCor$ | 5 |

*4.2. Evaluation Performance*

The IGD metric results of the benchmark test functions for the MOQSOA, three well-known classical algorithms (NSGA-II, MOEA/D, and MOPSO), and three state-of-the-art algorithms (IMMOEA, RVEA, and LMEA) are presented in Table 3, including mean values and standard deviations. The best values of the IGD metric are in bold. The Pareto front of each algorithm for ZDT3, ZDT4, DTLZ2, and DTLZ5 are shown in Figures 4–7.

From the statistical results of the IGD metrics in Table 3, it can be seen that the MOQSOA performed well on problems ZDT1, ZDT2, ZDT4, ZDT6, DTLZ1, DTLZ8, and DTLZ9, and achieved the best values for these test problems. On problems DTLZ2, DTLZ3, DTLZ5, and DTLZ6, although the best values of the indicators were obtained by the LMEA algorithm, the performance of MOQSOA was not significantly different from LMEA according to the results of the Wilcoxon signed-rank test, and was significantly better than the results obtained by the other algorithms on these problems.

LMEA performed better on problems ZDT3 and UF4-UF10, but MOQSOA also showed a good performance and the results obtained rank in the top three when comparing all algorithms. For the IGD metric, the MOQSOA obtained a mediocre performance only on problems DTLZ4 and UF1-UF3. The Pareto fronts of each algorithm in Figures 4–7 also showed the excellent performance of MOQSOA.

Comparing the performance of the IGD metrics for the two-objective and three-objective test problems, it can be seen that the MOQSOA outperformed NSGA-II, MOEA/D, MOPSO, IMMOEA, and RVEA on the two-objective test problem, and was basically equal to the LMEA algorithm. Additionally, for the three-objective test problem, the advantage over NSGA-II, MOEA/D, MOPSO, IMMOEA, and RVEA was obvious, but the algorithm was still slightly inferior to LMEA.

The Spacing metric results obtained for each algorithm on the benchmark test functions are presented in Table 4, where mean values and standard deviations of the results have been tabulated. Additionally, the best values of the Spacing metric for each test problem are shown in bold.

**Table 3.** IGD metric results.

| Function | Metrics | NSGAII | MOEA/D | MOPSO | IMMOEA | RVEA | LMEA | MOQSOA |
|---|---|---|---|---|---|---|---|---|
| ZDT1 | Average | $4.8116 \times 10^{-3}$ (−) | $4.6968 \times 10^{-3}$ (−) | $4.8735 \times 10^{-3}$ (−) | $7.9935 \times 10^{-3}$ (−) | $5.5528 \times 10^{-3}$ (−) | $5.1070 \times 10^{-3}$ (−) | $\mathbf{4.1101 \times 10^{-3}}$ |
| | Std | $2.06 \times 10^{-4}$ | $3.14 \times 10^{-4}$ | $1.77 \times 10^{-4}$ | $1.12 \times 10^{-4}$ | $9.19 \times 10^{-4}$ | $5.00 \times 10^{-4}$ | $7.28 \times 10^{-5}$ |
| ZDT2 | Average | $4.8719 \times 10^{-3}$ (−) | $5.4469 \times 10^{-3}$ (−) | $5.2469 \times 10^{-3}$ (−) | $1.1104 \times 10^{-2}$ (−) | $7.2369 \times 10^{-3}$ (−) | $4.6753 \times 10^{-3}$ (−) | $\mathbf{3.8136 \times 10^{-3}}$ |
| | Std | $1.81 \times 10^{-4}$ | $4.94 \times 10^{-4}$ | $3.24 \times 10^{-4}$ | $1.67 \times 10^{-4}$ | $1.74 \times 10^{-3}$ | $1.51 \times 10^{-4}$ | $5.67 \times 10^{-5}$ |
| ZDT3 | Average | $\mathbf{5.2780 \times 10^{-3}}$ (+) | $1.4329 \times 10^{-2}$ (−) | $5.5320 \times 10^{-3}$ (+) | $1.2438 \times 10^{-2}$ (−) | $7.9570 \times 10^{-3}$ (−) | $1.0940 \times 10^{-2}$ (−) | $6.4169 \times 10^{-3}$ |
| | Std | $2.39 \times 10^{-4}$ | $2.67 \times 10^{-3}$ | $2.12 \times 10^{-4}$ | $5.25 \times 10^{-4}$ | $1.88 \times 10^{-4}$ | $2.69 \times 10^{-3}$ | $1.39 \times 10^{-4}$ |
| ZDT4 | Average | $5.1081 \times 10^{-3}$ (−) | $7.2439 \times 10^{-3}$ (−) | $4.8082 \times 10^{-3}$ (−) | $4.8076 \times 10^{-3}$ (−) | $5.4592 \times 10^{-3}$ (−) | $4.7664 \times 10^{-3}$ (−) | $\mathbf{4.1890 \times 10^{-3}}$ |
| | Std | $6.53 \times 10^{-4}$ | $1.69 \times 10^{-3}$ | $3.00 \times 10^{-4}$ | $9.77 \times 10^{-5}$ | $1.44 \times 10^{-3}$ | $3.22 \times 10^{-4}$ | $2.64 \times 10^{-4}$ |
| ZDT6 | Average | $3.6160 \times 10^{-3}$ (−) | $4.3208 \times 10^{-3}$ (−) | $4.2740 \times 10^{-3}$ (−) | $9.4301 \times 10^{-1}$ (−) | $3.3918 \times 10^{-3}$ (−) | $3.3078 \times 10^{-3}$ (−) | $\mathbf{3.0046 \times 10^{-3}}$ |
| | Std | $1.20 \times 10^{-4}$ | $3.92 \times 10^{-3}$ | $2.73 \times 10^{-4}$ | $4.10 \times 10^{-2}$ | $2.25 \times 10^{-4}$ | $3.60 \times 10^{-5}$ | $2.18 \times 10^{-4}$ |
| DTLZ1 | Average | $2.7850 \times 10^{-2}$ (−) | $2.0565 \times 10^{-2}$ (=) | $2.7105 \times 10^{-2}$ (−) | $1.2940$ (−) | $2.0558 \times 10^{-2}$ (=) | $2.1025 \times 10^{-2}$ (=) | $\mathbf{2.0557 \times 10^{-2}}$ |
| | Std | $2.03 \times 10^{-3}$ | $1.07 \times 10^{-5}$ | $5.59 \times 10^{-4}$ | $6.58 \times 10^{-2}$ | $6.79 \times 10^{-5}$ | $2.55 \times 10^{-4}$ | $1.92 \times 10^{-4}$ |
| DTLZ2 | Average | $6.8830 \times 10^{-2}$ (−) | $5.4464 \times 10^{-2}$ (=) | $6.9818 \times 10^{-2}$ (−) | $7.9795 \times 10^{-2}$ (−) | $5.4465 \times 10^{-2}$ (=) | $\mathbf{5.3844 \times 10^{-2}}$ (=) | $5.4465 \times 10^{-2}$ |
| | Std | $3.19 \times 10^{-3}$ | $1.65 \times 10^{-5}$ | $3.22 \times 10^{-3}$ | $3.65 \times 10^{-3}$ | $1.28 \times 10^{-5}$ | $2.76 \times 10^{-4}$ | $1.34 \times 10^{-4}$ |
| DTLZ3 | Average | $6.297 \times 10^{-2}$ (−) | $5.5516 \times 10^{-2}$ (=) | $3.8831 \times 10^{-1}$ (−) | $2.8865$ (−) | $5.5358 \times 10^{-2}$ (=) | $\mathbf{5.4197 \times 10^{-2}}$ (=) | $5.4711 \times 10^{-2}$ |
| | Std | $4.49 \times 10^{-3}$ | $1.32 \times 10^{-3}$ | $5.40 \times 10^{-1}$ | $5.21 \times 10^{-1}$ | $1.15 \times 10^{-3}$ | $4.76 \times 10^{-4}$ | $1.23 \times 10^{-4}$ |
| DTLZ4 | Average | $6.7988 \times 10^{-2}$ (+) | $\mathbf{5.4464 \times 10^{-2}}$ (+) | $7.1277 \times 10^{-2}$ (+) | $7.7431 \times 10^{-2}$ (+) | $5.4465 \times 10^{-2}$ (+) | $9.6411 \times 10^{-2}$ (+) | $3.7873 \times 10^{-1}$ |
| | Std | $4.16 \times 10^{-3}$ | $4.87 \times 10^{-4}$ | $1.60 \times 10^{-3}$ | $3.48 \times 10^{-3}$ | $4.01 \times 10^{-4}$ | $7.37 \times 10^{-2}$ | $2.81 \times 10^{-1}$ |
| DTLZ5 | Average | $5.4096 \times 10^{-3}$ (=) | $3.3904 \times 10^{-2}$ (−) | $6.3037 \times 10^{-3}$ (−) | $2.0947 \times 10^{-2}$ (−) | $6.2925 \times 10^{-2}$ (−) | $\mathbf{4.6503 \times 10^{-3}}$ (=) | $5.0756 \times 10^{-3}$ |
| | Std | $6.67 \times 10^{-5}$ | $1.63 \times 10^{-5}$ | $9.93 \times 10^{-5}$ | $3.35 \times 10^{-3}$ | $2.09 \times 10^{-3}$ | $5.79 \times 10^{-5}$ | $1.90 \times 10^{-4}$ |
| DTLZ6 | Average | $5.8399 \times 10^{-3}$ (−) | $3.3926 \times 10^{-2}$ (−) | $6.7322 \times 10^{-3}$ (−) | $3.9896$ (−) | $1.1591 \times 10^{-1}$ (−) | $\mathbf{4.4731 \times 10^{-3}}$ (=) | $4.9789 \times 10^{-3}$ |
| | Std | $6.36 \times 10^{-5}$ | $3.15 \times 10^{-5}$ | $8.78 \times 10^{-4}$ | $7.10 \times 10^{-2}$ | $6.35 \times 10^{-4}$ | $9.76 \times 10^{-5}$ | $3.38 \times 10^{-5}$ |
| DTLZ7 | Average | $8.0964 \times 10^{-2}$ (+) | $1.5431 \times 10^{-1}$ (=) | $9.0199 \times 10^{-2}$ (+) | $3.2745 \times 10^{-1}$ (−) | $1.0659 \times 10^{-1}$ (+) | $\mathbf{5.8854 \times 10^{-2}}$ (+) | $1.6107 \times 10^{-1}$ |
| | Std | $4.70 \times 10^{-3}$ | $2.15 \times 10^{-4}$ | $1.30 \times 10^{-2}$ | $5.03 \times 10^{-2}$ | $2.21 \times 10^{-3}$ | $4.52 \times 10^{-4}$ | $1.62 \times 10^{-1}$ |
| DTLZ8 | Average | $4.4234 \times 10^{-2}$ (=) | NaN | NaN | NaN | $5.8818 \times 10^{-2}$ (−) | NaN | $\mathbf{4.3927 \times 10^{-2}}$ |
| | Std | $3.83 \times 10^{-3}$ | NaN | NaN | NaN | $1.43 \times 10^{-3}$ | NaN | $2.44 \times 10^{-3}$ |
| DTLZ9 | Average | $5.9530 \times 10^{-3}$ (−) | NaN | NaN | $4.4744$ (−) | $2.6833 \times 10^{-2}$ (−) | NaN | $\mathbf{5.1492 \times 10^{-3}}$ |
| | Std | $4.08 \times 10^{-4}$ | NaN | NaN | $6.01 \times 10^{-2}$ | $1.19 \times 10^{-3}$ | NaN | $4.92 \times 10^{-4}$ |
| UF1 | Average | $9.7093 \times 10^{-2}$ (=) | $2.6021 \times 10^{-1}$ (−) | $7.8469 \times 10^{-2}$ (+) | $6.7741 \times 10^{-2}$ (+) | $8.2189 \times 10^{-2}$ (+) | $\mathbf{1.6381 \times 10^{-2}}$ (+) | $1.4502 \times 10^{-1}$ |
| | Std | $3.41 \times 10^{-3}$ | $1.08 \times 10^{-1}$ | $7.31 \times 10^{-2}$ | $6.25 \times 10^{-3}$ | $5.02 \times 10^{-3}$ | $3.87 \times 10^{-3}$ | $4.59 \times 10^{-2}$ |
| UF2 | Average | $3.2343 \times 10^{-2}$ (+) | $8.2332 \times 10^{-2}$ (−) | $2.3067 \times 10^{-2}$ (+) | $5.4120 \times 10^{-2}$ (=) | $7.2716 \times 10^{-2}$ (−) | $\mathbf{1.5039 \times 10^{-2}}$ (+) | $5.3761 \times 10^{-2}$ |
| | Std | $3.81 \times 10^{-3}$ | $4.43 \times 10^{-2}$ | $3.00 \times 10^{-3}$ | $3.78 \times 10^{-2}$ | $7.74 \times 10^{-3}$ | $1.34 \times 10^{-3}$ | $2.03 \times 10^{-2}$ |
| UF3 | Average | $1.8641 \times 10^{-1}$ (+) | $3.1030 \times 10^{-1}$ (=) | $1.1567 \times 10^{-1}$ (+) | $\mathbf{1.1030 \times 10^{-1}}$ (+) | $3.1836 \times 10^{-1}$ (=) | $1.6484 \times 10^{-1}$ (+) | $2.8865 \times 10^{-1}$ |
| | Std | $1.19 \times 10^{-2}$ | $4.77 \times 10^{-2}$ | $2.14 \times 10^{-2}$ | $1.60 \times 10^{-2}$ | $2.26 \times 10^{-3}$ | $5.07 \times 10^{-3}$ | $1.40 \times 10^{-2}$ |

**Table 3.** *Cont.*

| Function | Metrics | NSGAII | MOEA/D | MOPSO | IMMOEA | RVEA | LMEA | MOQSOA |
|---|---|---|---|---|---|---|---|---|
| UF4 | Average | $4.9113 \times 10^{-2}$ (=) | $8.3692 \times 10^{-2}$ (−) | $4.5962 \times 10^{-2}$ (=) | $6.6832 \times 10^{-2}$ (−) | $9.5207 \times 10^{-2}$ (−) | $\mathbf{3.7822 \times 10^{-2}}$ (+) | $4.6945 \times 10^{-2}$ |
| | Std | $1.75 \times 10^{-3}$ | $3.42 \times 10^{-3}$ | $2.79 \times 10^{-3}$ | $4.24 \times 10^{-3}$ | $2.04 \times 10^{-3}$ | $5.38 \times 10^{-4}$ | $1.89 \times 10^{-3}$ |
| UF5 | Average | $3.9011 \times 10^{-1}$ (−) | $5.8082 \times 10^{-1}$ (−) | $6.7952 \times 10^{-1}$ (−) | $6.6328 \times 10^{-1}$ (−) | $3.4518 \times 10^{-1}$ (=) | $\mathbf{2.1331 \times 10^{-1}}$ (+) | $3.2603 \times 10^{-1}$ |
| | Std | $1.19 \times 10^{-1}$ | $8.90 \times 10^{-2}$ | $1.64 \times 10^{-1}$ | $9.79 \times 10^{-2}$ | $8.50 \times 10^{-2}$ | $3.20 \times 10^{-2}$ | $9.33 \times 10^{-2}$ |
| UF6 | Average | $\mathbf{1.2527 \times 10^{-1}}$ (+) | $4.9777 \times 10^{-1}$ (−) | $4.3455 \times 10^{-1}$ | $2.6280 \times 10^{-1}$ (=) | $1.2725 \times 10^{-1}$ (+) | $3.1444 \times 10^{-1}$ (−) | $2.6237 \times 10^{-1}$ |
| | Std | $1.25 \times 10^{-2}$ | $4.34 \times 10^{-1}$ | $8.01 \times 10^{-2}$ | $1.39 \times 10^{-1}$ | $9.55 \times 10^{-3}$ | $1.26 \times 10^{-2}$ | $1.35 \times 10^{-1}$ |
| UF7 | Average | $1.7075 \times 10^{-1}$ (=) | $4.3857 \times 10^{-1}$ (−) | $\mathbf{6.2768 \times 10^{-2}}$ (+) | $1.5326 \times 10^{-1}$ (=) | $1.3181 \times 10^{-1}$ (=) | $1.1450 \times 10^{-1}$ (=) | $1.4492 \times 10^{-1}$ |
| | Std | $1.56 \times 10^{-1}$ | $1.87 \times 10^{-1}$ | $7.63 \times 10^{-2}$ | $1.55 \times 10^{-1}$ | $1.70 \times 10^{-1}$ | $6.78 \times 10^{-2}$ | $1.32 \times 10^{-1}$ |
| UF8 | Average | $2.7066 \times 10^{-1}$ (−) | $3.2370 \times 10^{-1}$ (−) | $2.6221 \times 10^{-1}$ (−) | $2.7670 \times 10^{-1}$ (−) | $3.3376 \times 10^{-1}$ (−) | $\mathbf{1.5603 \times 10^{-1}}$ (+) | $2.1688 \times 10^{-1}$ |
| | Std | $7.49 \times 10^{-2}$ | $3.07 \times 10^{-2}$ | $7.18 \times 10^{-2}$ | $3.10 \times 10^{-3}$ | $5.66 \times 10^{-3}$ | $1.60 \times 10^{-2}$ | $7.29 \times 10^{-2}$ |
| UF9 | Average | $2.6615 \times 10^{-1}$ (=) | $3.4263 \times 10^{-1}$ (−) | $2.8580 \times 10^{-1}$ (−) | $3.0671 \times 10^{-1}$ (−) | $3.6412 \times 10^{-1}$ (−) | $\mathbf{9.0845 \times 10^{-2}}$ (+) | $2.3198 \times 10^{-1}$ |
| | Std | $9.57 \times 10^{-2}$ | $8.27 \times 10^{-3}$ | $2.09 \times 10^{-2}$ | $1.22 \times 10^{-1}$ | $1.89 \times 10^{-2}$ | $3.04 \times 10^{-2}$ | $5.95 \times 10^{-2}$ |
| UF10 | Average | $4.3683 \times 10^{-1}$ (=) | $7.9220 \times 10^{-1}$ (−) | $5.5064 \times 10^{-1}$ (−) | $\mathbf{2.9879 \times 10^{-1}}$ (+) | $6.5234 \times 10^{-1}$ (−) | $4.6990 \times 10^{-1}$ (=) | $4.3228 \times 10^{-1}$ |
| | Std | $5.88 \times 10^{-2}$ | $1.35 \times 10^{-1}$ | $3.59 \times 10^{-2}$ | $5.14 \times 10^{-3}$ | $2.11 \times 10^{-1}$ | $3.90 \times 10^{-2}$ | $1.41 \times 10^{-1}$ |
| +/ − / = | | 6/11/7 | 1/16/5 | 7/14/1 | 4/16/3 | 4/14/6 | 9/6/7 | |

**Table 4.** Spacing metric results.

| Function | Metrics | NSGAII | MOEA/D | MOPSO | IMMOEA | RVEA | LMEA | MOQSOA |
|---|---|---|---|---|---|---|---|---|
| ZDT1 | Average | $6.8090 \times 10^{-3}$ (−) | $\mathbf{5.3019 \times 10^{-3}}$ (−) | $7.7359 \times 10^{-3}$ (−) | $1.4474 \times 10^{-2}$ (−) | $9.7893 \times 10^{-3}$ (−) | $1.3206 \times 10^{-2}$ (−) | $5.3225 \times 10^{-3}$ |
| | Std | $5.22 \times 10^{-4}$ | $4.93 \times 10^{-4}$ | $7.03 \times 10^{-4}$ | $6.41 \times 10^{-3}$ | $5.08 \times 10^{-4}$ | $3.48 \times 10^{-3}$ | $6.57 \times 10^{-4}$ |
| ZDT2 | Average | $7.5232 \times 10^{-3}$ (−) | $5.0442 \times 10^{-3}$ (=) | $7.7292 \times 10^{-3}$ (−) | $9.2439 \times 10^{-3}$ (−) | $7.1899 \times 10^{-3}$ (−) | $5.0437 \times 10^{-3}$ (=) | $\mathbf{4.4243 \times 10^{-3}}$ |
| | Std | $8.84 \times 10^{-4}$ | $5.62 \times 10^{-4}$ | $4.86 \times 10^{-4}$ | $4.41 \times 10^{-4}$ | $2.32 \times 10^{-3}$ | $1.07 \times 10^{-3}$ | $1.27 \times 10^{-4}$ |
| ZDT3 | Average | $\mathbf{7.4689 \times 10^{-3}}$ (+) | $1.9846 \times 10^{-2}$ (=) | $7.8757 \times 10^{-3}$ (+) | $3.3663 \times 10^{-2}$ (−) | $1.1866 \times 10^{-2}$ (+) | $1.2563 \times 10^{-2}$ (=) | $1.3825 \times 10^{-2}$ |
| | Std | $6.95 \times 10^{-4}$ | $2.08 \times 10^{-3}$ | $7.14 \times 10^{-4}$ | $9.87 \times 10^{-4}$ | $9.30 \times 10^{-4}$ | $2.53 \times 10^{-3}$ | $3.47 \times 10^{-5}$ |
| ZDT4 | Average | $7.1956 \times 10^{-3}$ (+) | $\mathbf{5.6232 \times 10^{-3}}$ (+) | $7.1330 \times 10^{-3}$ (+) | $7.4260 \times 10^{-3}$ (+) | $9.7327 \times 10^{-3}$ (+) | $1.4311 \times 10^{-2}$ (−) | $1.0226 \times 10^{-2}$ |
| | Std | $5.70 \times 10^{-4}$ | $1.18 \times 10^{-3}$ | $5.23 \times 10^{-4}$ | $5.01 \times 10^{-4}$ | $3.54 \times 10^{-4}$ | $1.59 \times 10^{-3}$ | $6.01 \times 10^{-4}$ |
| ZDT6 | Average | $5.6569 \times 10^{-3}$ (−) | $3.2179 \times 10^{-3}$ (=) | $7.6746 \times 10^{-3}$ (−) | $5.3819 \times 10^{-2}$ (−) | $2.3782 \times 10^{-3}$ (−) | $3.7229 \times 10^{-3}$ (−) | $\mathbf{2.1262 \times 10^{-3}}$ |
| | Std | $5.31 \times 10^{-4}$ | $2.58 \times 10^{-4}$ | $4.97 \times 10^{-4}$ | $1.60 \times 10^{-2}$ | $7.91 \times 10^{-5}$ | $4.36 \times 10^{-4}$ | $3.36 \times 10^{-4}$ |
| DTLZ1 | Average | $2.1135 \times 10^{-2}$ (+) | $\mathbf{3.7899 \times 10^{-5}}$ (+) | $2.2926 \times 10^{-2}$ (+) | $2.2280$ (−) | $1.6424 \times 10^{-4}$ (+) | $1.8593 \times 10^{-2}$ (+) | $3.1105 \times 10^{-2}$ |
| | Std | $1.33 \times 10^{-3}$ | $7.94 \times 10^{-5}$ | $1.98 \times 10^{-3}$ | $8.39 \times 10^{-1}$ | $6.07 \times 10^{-4}$ | $1.74 \times 10^{-3}$ | $1.15 \times 10^{-3}$ |
| DTLZ2 | Average | $5.7049 \times 10^{-2}$ (+) | $5.7179 \times 10^{-2}$ (+) | $6.0138 \times 10^{-2}$ (+) | $8.6819 \times 10^{-2}$ (=) | $5.7164 \times 10^{-2}$ (+) | $\mathbf{2.7904 \times 10^{-2}}$ (+) | $8.4078 \times 10^{-2}$ |
| | Std | $4.56 \times 10^{-3}$ | $4.49 \times 10^{-5}$ | $6.81 \times 10^{-3}$ | $3.42 \times 10^{-3}$ | $6.08 \times 10^{-5}$ | $2.99 \times 10^{-3}$ | $3.39 \times 10^{-3}$ |

**Table 4.** *Cont.*

| Function | Metrics | NSGAII | MOEA/D | MOPSO | IMMOEA | RVEA | LMEA | MOQSOA |
|---|---|---|---|---|---|---|---|---|
| DTLZ3 | Average | $1.4015 \times 10^{-1}$ (−) | $5.6364 \times 10^{-2}$ (+) | $8.4753 \times 10^{-2}$ (=) | $5.5285$ (−) | $5.4953 \times 10^{-2}$ (+) | $\mathbf{3.0975 \times 10^{-2}}$ (+) | $8.3840 \times 10^{-2}$ |
| | Std | $1.47 \times 10^{-1}$ | $1.51 \times 10^{-3}$ | $3.47 \times 10^{-2}$ | $1.63$ | $3.93 \times 10^{-3}$ | $1.06 \times 10^{-3}$ | $2.00 \times 10^{-3}$ |
| DTLZ4 | Average | $5.4198 \times 10^{-2}$ (−) | $5.7166 \times 10^{-2}$ (−) | $6.3045 \times 10^{-2}$ (−) | $7.1517 \times 10^{-2}$ (−) | $5.7146 \times 10^{-2}$ (−) | $5.8702 \times 10^{-2}$ (−) | $\mathbf{3.2762 \times 10^{-2}}$ |
| | Std | $4.46 \times 10^{-3}$ | $1.01 \times 10^{-4}$ | $4.18 \times 10^{-3}$ | $5.65 \times 10^{-3}$ | $2.37 \times 10^{-4}$ | $5.31 \times 10^{-2}$ | $4.08 \times 10^{-2}$ |
| DTLZ5 | Average | $8.8231 \times 10^{-3}$ (+) | $1.3776 \times 10^{-2}$ (=) | $1.1250 \times 10^{-2}$ (=) | $5.2354 \times 10^{-2}$ (−) | $1.2206 \times 10^{-1}$ (−) | $\mathbf{7.9189 \times 10^{-3}}$ (+) | $1.2907 \times 10^{-2}$ |
| | Std | $1.53 \times 10^{-4}$ | $8.58 \times 10^{-5}$ | $2.95 \times 10^{-4}$ | $2.41 \times 10^{-3}$ | $1.43 \times 10^{-2}$ | $4.14 \times 10^{-4}$ | $6.24 \times 10^{-4}$ |
| DTLZ6 | Average | $1.1714 \times 10^{-2}$ (=) | $1.2549 \times 10^{-2}$ (=) | $1.1354 \times 10^{-2}$ (=) | $4.6209 \times 10^{-1}$ (−) | $1.0722 \times 10^{-1}$ (−) | $\mathbf{7.0155 \times 10^{-3}}$ (+) | $1.2237 \times 10^{-2}$ |
| | Std | $5.38 \times 10^{-4}$ | $6.61 \times 10^{-5}$ | $4.92 \times 10^{-5}$ | $9.05 \times 10^{-2}$ | $1.32 \times 10^{-3}$ | $1.25 \times 10^{-3}$ | $2.05 \times 10^{-4}$ |
| DTLZ7 | Average | $6.3775 \times 10^{-2}$ (+) | $1.9627 \times 10^{-1}$ (−) | $7.6841 \times 10^{-2}$ (=) | $2.5247 \times 10^{-1}$ (−) | $1.1524 \times 10^{-1}$ (−) | $\mathbf{5.9592 \times 10^{-2}}$ (+) | $8.2865 \times 10^{-2}$ |
| | Std | $9.14 \times 10^{-3}$ | $9.65 \times 10^{-4}$ | $4.31 \times 10^{-3}$ | $3.51 \times 10^{-2}$ | $1.25 \times 10^{-3}$ | $7.30 \times 10^{-3}$ | $2.81 \times 10^{-2}$ |
| DTLZ8 | Average | $3.6512 \times 10^{-2}$ (=) | NaN | NaN | NaN | $\mathbf{3.3160 \times 10^{-2}}$ (=) | NaN | $3.8990 \times 10^{-2}$ |
| | Std | $9.12 \times 10^{-3}$ | NaN | NaN | NaN | $5.54 \times 10^{-3}$ | NaN | $4.24 \times 10^{-3}$ |
| DTLZ9 | Average | $8.4354 \times 10^{-3}$ (−) | NaN | NaN | $8.7077 \times 10^{-2}$ (−) | $3.0606 \times 10^{-2}$ | NaN | $\mathbf{7.1642 \times 10^{-3}}$ |
| | Std | $6.68 \times 10^{-4}$ | NaN | NaN | $2.46 \times 10^{-2}$ | $5.96 \times 10^{-3}$ | NaN | $1.29 \times 10^{-3}$ |
| UF1 | Average | $\mathbf{2.3718 \times 10^{-3}}$ (=) | $3.5533 \times 10^{-3}$ (−) | $1.5285 \times 10^{-2}$ (−) | $2.4196 \times 10^{-2}$ (−) | $2.3252 \times 10^{-2}$ (−) | $1.6681 \times 10^{-2}$ (−) | $2.4022 \times 10^{-3}$ |
| | Std | $2.35 \times 10^{-3}$ | $5.10 \times 10^{-3}$ | $2.12 \times 10^{-2}$ | $2.77 \times 10^{-2}$ | $6.39 \times 10^{-3}$ | $3.62 \times 10^{-3}$ | $1.53 \times 10^{-3}$ |
| UF2 | Average | $\mathbf{5.2999 \times 10^{-3}}$ (+) | $8.4767 \times 10^{-3}$ (+) | $5.9590 \times 10^{-3}$ (+) | $1.0650 \times 10^{-2}$ (+) | $1.1906 \times 10^{-2}$ (=) | $1.3808 \times 10^{-2}$ (=) | $1.4313 \times 10^{-2}$ |
| | Std | $7.03 \times 10^{-4}$ | $5.03 \times 10^{-3}$ | $3.62 \times 10^{-4}$ | $8.10 \times 10^{-4}$ | $8.61 \times 10^{-4}$ | $4.06 \times 10^{-3}$ | $1.12 \times 10^{-2}$ |
| UF3 | Average | $2.0528 \times 10^{-2}$ (−) | $2.5562 \times 10^{-3}$ (+) | $1.1321 \times 10^{-2}$ (−) | $1.0894 \times 10^{-2}$ (−) | $\mathbf{6.5446 \times 10^{-4}}$ (+) | $4.3037 \times 10^{-2}$ (−) | $4.7750 \times 10^{-3}$ |
| | Std | $1.75 \times 10^{-2}$ | $5.03 \times 10^{-3}$ | $1.11 \times 10^{-2}$ | $1.92 \times 10^{-3}$ | $7.68 \times 10^{-4}$ | $1.45 \times 10^{-2}$ | $6.76 \times 10^{-3}$ |
| UF4 | Average | $\mathbf{6.6588 \times 10^{-3}}$ (=) | $9.0938 \times 10^{-3}$ (−) | $7.2809 \times 10^{-3}$ (=) | $1.1696 \times 10^{-2}$ (−) | $1.8483 \times 10^{-2}$ (−) | $1.0004 \times 10^{-2}$ (−) | $6.9043 \times 10^{-3}$ |
| | Std | $8.69 \times 10^{-4}$ | $1.51 \times 10^{-3}$ | $5.88 \times 10^{-4}$ | $1.18 \times 10^{-3}$ | $5.22 \times 10^{-3}$ | $1.44 \times 10^{-3}$ | $6.55 \times 10^{-4}$ |
| UF5 | Average | $2.7938 \times 10^{-2}$ (=) | $\mathbf{4.5153 \times 10^{-4}}$ (+) | $1.4962 \times 10^{-2}$ (+) | $6.5039 \times 10^{-2}$ (−) | $6.7016 \times 10^{-2}$ (−) | $1.4595 \times 10^{-1}$ (−) | $2.7739 \times 10^{-2}$ |
| | Std | $2.15 \times 10^{-2}$ | $9.05 \times 10^{-4}$ | $1.29 \times 10^{-2}$ | $4.55 \times 10^{-2}$ | $4.03 \times 10^{-2}$ | $8.47 \times 10^{-2}$ | $2.45 \times 10^{-2}$ |
| UF6 | Average | $6.5459 \times 10^{-2}$ (−) | $5.4275 \times 10^{-2}$ (−) | $1.0577 \times 10^{-2}$ (−) | $2.4009 \times 10^{-2}$ (−) | $2.3187 \times 10^{-1}$ (−) | $9.4877 \times 10^{-2}$ (−) | $\mathbf{5.8230 \times 10^{-3}}$ |
| | Std | $6.07 \times 10^{-2}$ | $7.86 \times 10^{-2}$ | $1.83 \times 10^{-2}$ | $1.45 \times 10^{-2}$ | $2.95 \times 10^{-1}$ | $4.76 \times 10^{-2}$ | $5.38 \times 10^{-3}$ |
| UF7 | Average | $\mathbf{2.5451 \times 10^{-3}}$ (+) | $4.0496 \times 10^{-3}$ (+) | $7.8085 \times 10^{-3}$ (−) | $1.2477 \times 10^{-2}$ (−) | $1.6111 \times 10^{-2}$ (−) | $3.7577 \times 10^{-2}$ (−) | $6.0753 \times 10^{-3}$ |
| | Std | $1.89 \times 10^{-3}$ | $7.63 \times 10^{-3}$ | $5.71 \times 10^{-3}$ | $2.76 \times 10^{-3}$ | $8.23 \times 10^{-3}$ | $3.28 \times 10^{-2}$ | $6.67 \times 10^{-3}$ |
| UF8 | Average | $1.3926 \times 10^{-1}$ (=) | $2.1788 \times 10^{-1}$ (−) | $1.0950 \times 10^{-1}$ (=) | $1.5689 \times 10^{-1}$ (−) | $2.6998 \times 10^{-1}$ (−) | $\mathbf{6.3257 \times 10^{-2}}$ (+) | $1.2436 \times 10^{-1}$ |
| | Std | $1.65 \times 10^{-2}$ | $7.52 \times 10^{-2}$ | $2.95 \times 10^{-2}$ | $3.80 \times 10^{-2}$ | $5.11 \times 10^{-2}$ | $5.47 \times 10^{-3}$ | $3.87 \times 10^{-2}$ |
| UF9 | Average | $1.0959 \times 10^{-1}$ (−) | $8.7028 \times 10^{-2}$ (−) | $9.9912 \times 10^{-2}$ (−) | $6.0917 \times 10^{-1}$ (−) | $1.2816 \times 10^{-1}$ (−) | $\mathbf{6.4226 \times 10^{-2}}$ (+) | $7.6352 \times 10^{-2}$ |
| | Std | $1.96 \times 10^{-2}$ | $1.70 \times 10^{-2}$ | $2.67 \times 10^{-2}$ | $1.72 \times 10^{-1}$ | $5.27 \times 10^{-3}$ | $6.94 \times 10^{-3}$ | $1.44 \times 10^{-2}$ |
| UF10 | Average | $2.1199 \times 10^{-1}$ (−) | $\mathbf{2.8860 \times 10^{-3}}$ (+) | $1.7352 \times 10^{-1}$ (−) | $3.2344 \times 10^{-1}$ (−) | $5.5619 \times 10^{-1}$ (−) | $9.4595 \times 10^{-2}$ (+) | $1.3956 \times 10^{-1}$ |
| | Std | $9.62 \times 10^{-2}$ | $2.66 \times 10^{-3}$ | $2.98 \times 10^{-2}$ | $3.67 \times 10^{-2}$ | $4.79 \times 10^{-1}$ | $7.18 \times 10^{-2}$ | $1.61 \times 10^{-1}$ |
| +/ − / = | | 8/10/6 | 9/8/5 | 6/10/6 | 2/20/1 | 6/16/2 | 9/10/3 | |

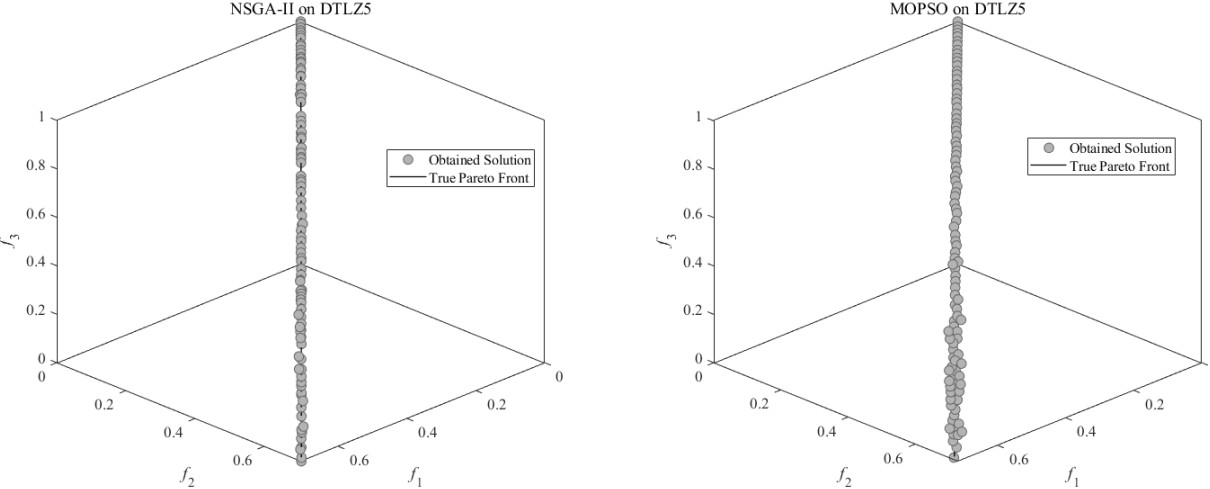

**Figure 4.** Pareto fronts of each algorithm for ZDT4.

**Figure 5.** *Cont.*

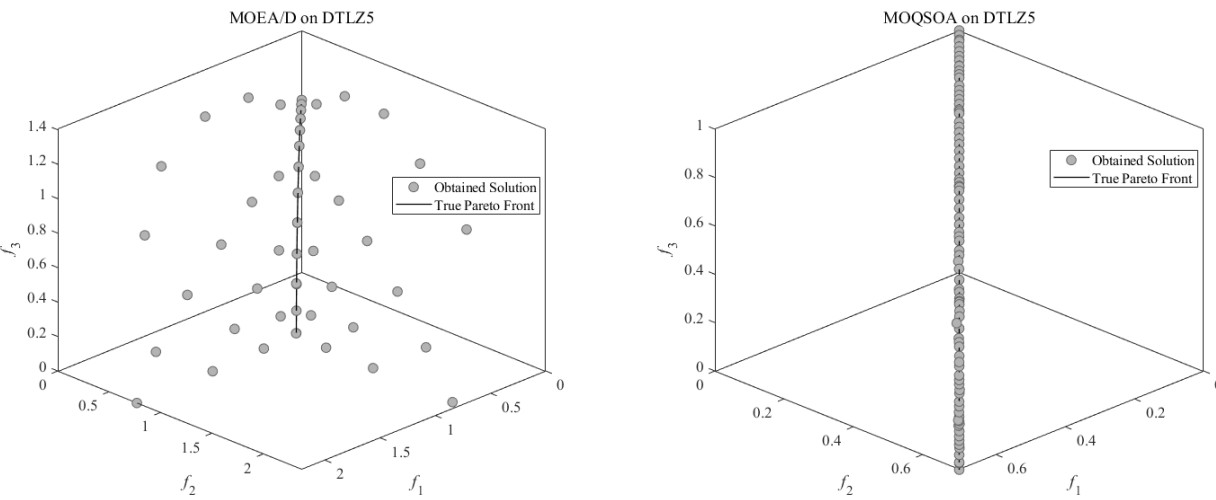

**Figure 5.** Pareto fronts of each algorithm for DTLZ5.

**Figure 6.** Pareto fronts of each algorithm for ZDT3.

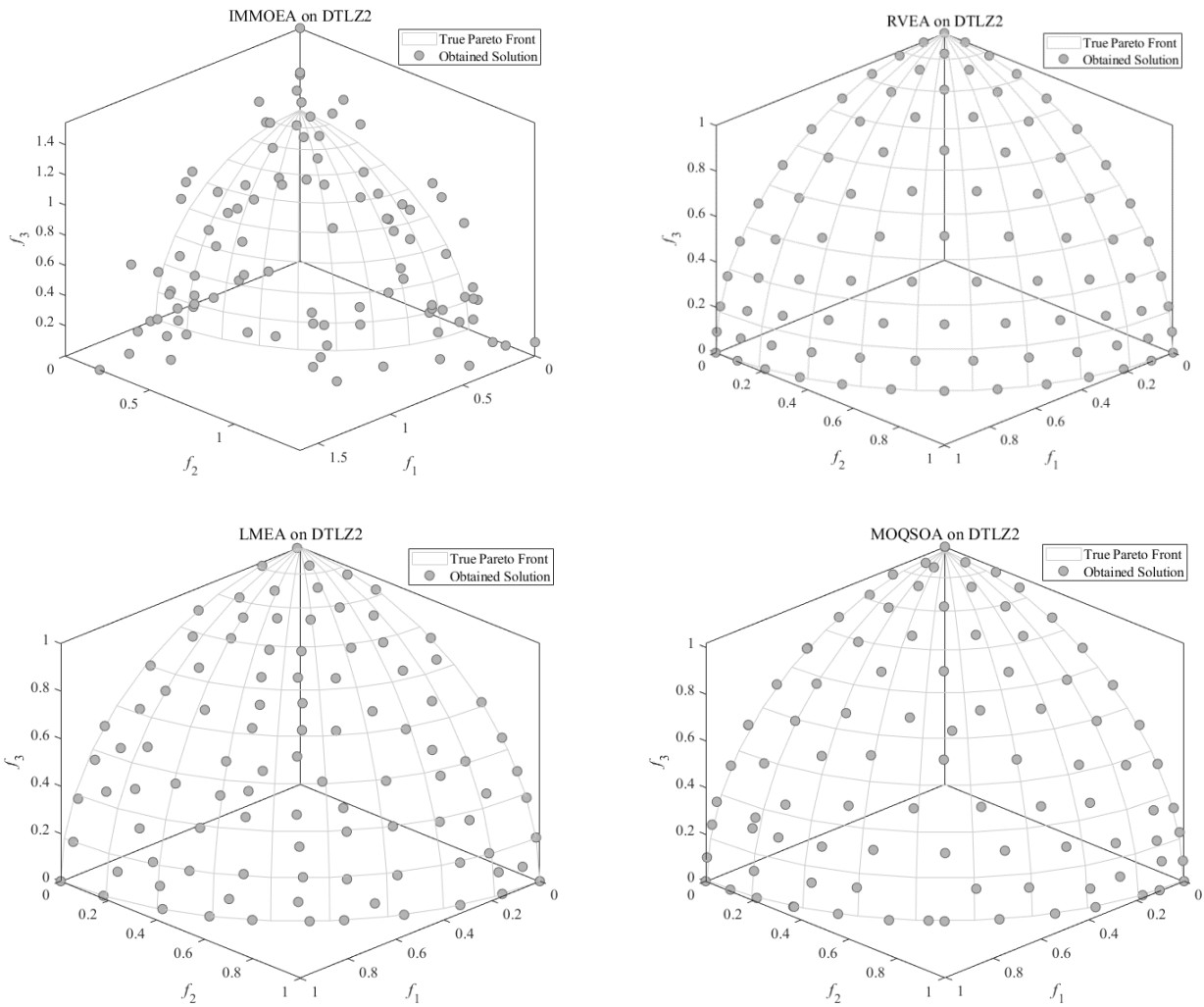

**Figure 7.** Pareto fronts of each algorithm for DTLZ2.

Compared to the other algorithms in Table 4, the MOQSOA exhibited a better performance with the Spacing metric. Specifically, for problems ZDT2, ZDT6, DTLZ4, DTLZ9, and UF6, the MOQSOA achieved the best values. For problems ZDT1, DTLZ8, UF1, and UF4, although the best values of the indicators were obtained by NSGA-II, MOEA/D, and RVEA, the results obtained by the MOQSOA did not differ significantly from the optimal values according to the Wilcoxon signed-rank test, which reflects the advantage of the proposed algorithm in the distribution of the solutions. In addition, for problems DTLZ5-DTLZ7, UF3, UF5, and UF7-UF10, the LMEA and MOEA/D algorithms performed better, but the MOQSOA also showed a good performance and the obtained results ranked in the top three when comparing all algorithms. For the Spacing metric, the MOQSOA only underperformed on problems ZDT3-ZDT4, DTLZ1-DTLZ3, and UF2.

When comparing the performance of the Spacing metric for the two-objective and three-objective test problems, it can be seen that the MOQSOA performed better in the distribution of the solutions than MOPSO, IMMOEA, and RVEA on the two-objective test problem, and was basically equal to NSGA-II and MOEA/D. Additionally, for the three-objective test problem, the advantage over MOPSO, IMMOEA and RVEA was more obvious, but the algorithm was still slightly inferior to LMEA.

Through statistics and the analysis of the experimental results, it has been proven that the proposed MOQSOA has good performance in dealing with multi-objective optimization problems. The MOQSOA significantly improved the convergence and distribution of solu-

tions in the test problems compared to the other multi-objective optimization algorithms. Specifically, the convergence of the MOQSOA was significantly enhanced compared to the classical multi-objective optimization algorithms, and the distribution of the solutions was improved compared to the novel multi-objective optimization algorithm. In addition, the MOQSOA can balance the convergence and the distribution of solutions well.

### 4.3. The Influence of Strategies

#### 4.3.1. The Influence of Real-Coded Quantum Representation

Inspired by the QEA, the proposed MOQSOA treats the current optimal solution as a linear superposition of two probabilistic states with real-coded quantum representation. Each seagull individual makes its own judgment on whether to accept the current optimal solution as the global optimum during the iterations. To demonstrate the effectiveness of this strategy, the proposed MOQSOA was compared with the MOSOA [24] that adopts the basic seagull optimization algorithm by IGD metrics and Spacing metrics on the test functions ZDT1, ZDT2, ZDT3, ZDT6, DTLZ4, DTLZ6, and UF6. The population size and the maximum capacity of the archive were set to 100, the maximum number of iterations was set to 1000, and each algorithm was implemented for 30 independent runs. The means and standard deviations are presented in Table 5. The Wilcoxon signed-rank test [60] was performed. The results are shown in Table 5 with a significance level of $\alpha = 0.05$, and "+", "−", and "=" indicate that the MOSOA is superior, inferior, or equal to the MOQSOA, respectively.

**Table 5.** Results influenced with real-coded quantum representation.

| Function | Metrics | MOSOA IGD | MOQSOA IGD | MOSOA Spacing | MOQSOA Spacing |
|---|---|---|---|---|---|
| ZDT1 | Average | $4.0025 \times 10^{-3}$ (=) | $4.1101 \times 10^{-3}$ | $5.1220 \times 10^{-3}$ (=) | $5.3225 \times 10^{-3}$ |
| | Std | $6.55 \times 10^{-5}$ | $7.28 \times 10^{-5}$ | $3.26 \times 10^{-4}$ | $6.57 \times 10^{-4}$ |
| ZDT2 | Average | $3.8531 \times 10^{-3}$ (=) | $3.8136 \times 10^{-3}$ | $4.7874 \times 10^{-3}$ (=) | $4.4243 \times 10^{-3}$ |
| | Std | $3.04 \times 10^{-5}$ | $5.67 \times 10^{-5}$ | $2.14 \times 10^{-4}$ | $1.27 \times 10^{-4}$ |
| ZDT3 | Average | $7.4483 \times 10^{-3}$ (−) | $6.4169 \times 10^{-3}$ | $1.4084 \times 10^{-2}$ (=) | $1.3825 \times 10^{-2}$ |
| | Std | $5.28 \times 10^{-4}$ | $1.39 \times 10^{-4}$ | $1.78 \times 10^{-4}$ | $3.47 \times 10^{-5}$ |
| ZDT6 | Average | $3.9728 \times 10^{-3}$ (−) | $3.0046 \times 10^{-3}$ | $2.2837 \times 10^{-3}$ (=) | $2.1262 \times 10^{-3}$ |
| | Std | $3.59 \times 10^{-4}$ | $2.18 \times 10^{-4}$ | $1.11 \times 10^{-5}$ | $3.36 \times 10^{-4}$ |
| DTLZ4 | Average | $5.1377 \times 10^{-1}$ (−) | $3.7873 \times 10^{-1}$ | $3.1023 \times 10^{-2}$ (=) | $3.2762 \times 10^{-2}$ |
| | Std | $4.46 \times 10^{-1}$ | $2.81 \times 10^{-1}$ | $4.54 \times 10^{-2}$ | $4.08 \times 10^{-2}$ |
| DTLZ6 | Average | $6.3906 \times 10^{-3}$ (−) | $4.9789 \times 10^{-3}$ | $1.2702 \times 10^{-2}$ (=) | $1.2237 \times 10^{-2}$ |
| | Std | $2.06 \times 10^{-5}$ | $3.38 \times 10^{-5}$ | $2.85 \times 10^{-4}$ | $2.05 \times 10^{-4}$ |
| UF6 | Average | $3.6369 \times 10^{-1}$ (−) | $2.6237 \times 10^{-1}$ | $8.1922 \times 10^{-3}$ (−) | $5.8230 \times 10^{-3}$ |
| | Std | $1.95 \times 10^{-1}$ | $1.35 \times 10^{-1}$ | $5.45 \times 10^{-3}$ | $5.38 \times 10^{-3}$ |
| +/ − / = | | 0/5/2 | | 0/1/6 | |

Based on the results, as shown in Table 5, it can be seen that for problems ZDT3, ZDT6, DTLZ4, DTLZ6, and UF6, the IGD values obtained by MOQSOA were better than those of the MOSOA. In contrast, for problems ZDT1 and ZDT2, the performance of the MOQSOA was not significantly different from the performance of the MOSOA. As for the Spacing metric, the MOQSOA did not differ significantly from the MOSOA on the majority of problems according to the Wilcoxon signed-rank test. The experimental results illustrate that, by adding real-coded quantum representation for current optimal solution, the MOQSOA is able to improve the convergence of the algorithm without affecting the distribution of the solutions, and shows better performance.

Generally, the real-coded quantum representation strategy helps to improve the algorithm in searching for global optimal solutions and identifying local optimal stagnation.

### 4.3.2. The Influence of Nonlinear Migration Operation

Instead of adopting the linear descent approach for the additional variable $A$ in the migration operation in the basic SOA, the proposed MOQSOA uses a nonlinearly varying variable $A$ to match the migration process of the actual seagull population better, as well as accelerate the convergence and improve the search accuracy of the algorithm. To demonstrate the effectiveness of this strategy, the proposed MOQSOA was compared with the MOQSOA that adopts the linear descent method of the control variable $A$ in the basic SOA (denoted as MOQSOA-LD) on test problems ZDT1, ZDT3, and DTLZ6. Additionally, the experiment metrics were the IGD metrics of the 200th, 500th, and 1000th generations. The population size and the maximum capacity of the archive were set to 100, the maximum number of iterations was set to 1000, and each algorithm was implemented for 30 independent runs. The means and standard deviations are presented in Table 6. The Wilcoxon signed-rank test [60] was performed. The results are shown in Table 5 with a significance level of $\alpha = 0.05$, and "+", "−", and "=" indicate that MOQSOA-LD is superior, inferior, or equal to MOQSOA, respectively.

**Table 6.** Results influenced with nonlinear migration operation.

| Function | Metrics | MOQSOA-LD 200th Iteration | MOQSOA 200th Iteration | MOQSOA-LD 500th Iteration | MOQSOA 500th Iteration | MOQSOA-LD 1000th Iteration | MOQSOA 1000th Iteration |
|---|---|---|---|---|---|---|---|
| ZDT1 | Average | $5.7792 \times 10^{-3}$ (−) | $4.1651 \times 10^{-3}$ | $3.8902 \times 10^{-3}$ (=) | $3.8901 \times 10^{-3}$ | $3.8882 \times 10^{-3}$ (=) | $3.8881 \times 10^{-3}$ |
| | Std | $3.70 \times 10^{-4}$ | $1.16 \times 10^{-4}$ | $8.27 \times 10^{-5}$ | $1.18 \times 10^{-6}$ | $5.25 \times 10^{-8}$ | $8.11 \times 10^{-8}$ |
| ZDT2 | Average | $9.7120 \times 10^{-3}$ (−) | $6.6238 \times 10^{-3}$ | $6.4306 \times 10^{-3}$ (=) | $6.4205 \times 10^{-3}$ | $6.4202 \times 10^{-3}$ (=) | $6.4162 \times 10^{-3}$ |
| | Std | $4.52 \times 10^{-3}$ | $1.61 \times 10^{-4}$ | $2.00 \times 10^{-5}$ | $1.07 \times 10^{-5}$ | $2.72 \times 10^{-6}$ | $7.51 \times 10^{-6}$ |
| DTLZ6 | Average | $5.0310 \times 10^{-3}$ (=) | $4.9659 \times 10^{-3}$ | $5.0066 \times 10^{-3}$ (=) | $4.9540 \times 10^{-3}$ | $4.9813 \times 10^{-3}$ (=) | $4.9534 \times 10^{-3}$ |
| | Std | $4.28 \times 10^{-5}$ | $7.84 \times 10^{-5}$ | $9.53 \times 10^{-5}$ | $3.38 \times 10^{-5}$ | $5.38 \times 10^{-5}$ | $8.62 \times 10^{-5}$ |
| +/ − / = | | 0/2/1 | | 0/0/3 | | 0/0/3 | |

Based on the results illustrated in Table 6, it can be seen that the MOQSOA generally performs better than MOQSOA-LD for problems ZDT1, ZDT3, and DTLZ6 at the 200th generation. Although the effect produced by the nonlinear migration strategy is no longer apparent in the later stage of iteration, this strategy can improve exploitation in the early stage of iteration.

As a summary, the abilities of exploitation and convergence are emphasized due to the employed real-coded quantum representation and nonlinear migration operation strategies, which help the proposed MOQSOA to obtain a better performance for different kinds of problems.

## 5. Conclusions

Multi-objective optimization algorithms need to balance convergence with distribution. However, many multi-objective optimization algorithms are prone to local optimization, leading to unbalanced convergence and distribution problems. In order to counterpoise the convergence and distribution of Pareto optimal solutions in MOPs, a multi-objective quantum-inspired seagull optimization algorithm, termed MOQSOA, was proposed in this paper. The proposed algorithm combined opposite-based learning, the migration and attacking behavior of seagulls, grid ranking, and the superposition principles of quantum computing. To obtain a better initialized population in the absence of a priori knowledge, an OBL mechanism was used to initialize the seagull population. Furthermore, it contained the nonlinear migration and attacking operations of the SOA. To maintain a better balance between exploitation and exploration when searching global optimal solutions, the proposed algorithm adapted the real-coded quantum representation of the current optimal solution and quantum rotation gate. In addition, the grid mechanism with GGR and GDR provided a criterion for leader selection and archive control. To evaluate the performance of the proposed algorithm in this paper, NSGA-II, MOEA/D, MOPSO, IMMOEA, RVEA, and LMEA were selected as comparative algorithms. The results of the

tests performed on the ZDT, DTLZ, and UF test suites demonstrated that the MOQSOA was able to enhance the distribution and convergence performance of MOPs.

The proposed MOQSOA showed effectiveness and efficiency in the MOP benchmark test problems. However, there is still a lot of potential future work that deserves to be studied in depth. One desirable future investigation is to solve specific real-life difficult engineering problems with the proposed algorithm, such as circuit designing, electronic component arrangement, cost optimization, automatic navigation, and sustainable energy systems. Additionally, it is worthy studying how to determine whether an optimal solution is positive or deceptive more scientifically. In addition, the potential capability of the MOQSOA to solve many objective optimization problems should be demonstrated. Moreover, it will be interesting to investigate how to use the principles of quantum computing in other multi-objective optimization algorithms.

**Author Contributions:** Conceptualization, Y.W., W.W. and I.A.; methodology, Y.W. and I.A.; software, Y.W. and I.A.; validation, Y.W.; formal analysis, Y.W.; investigation, Y.W.; resources, Y.W., W.W. and I.A.; data curation, Y.W.; writing—original draft preparation, Y.W.; writing—review and editing, Y.W., W.W., I.A. and E.T.-E.; visualization, Y.W.; supervision, W.W.; project administration, W.W.; funding acquisition, W.W. All authors have read and agreed to the published version of the manuscript.

**Funding:** This research was funded by the National Natural Science Foundation of China (No. 61873240) and Faculty of Engineering and Technology, Future University in Egypt, 440 New Cairo 11845, Egypt.

**Data Availability Statement:** The data presented in this study are openly available in PlatEMO at https://doi.org/10.1109/MCI.2017.2742868 (accessed on 23 May 2022), reference number [59].

**Acknowledgments:** The authors would like to thank the anonymous reviewers for their constructive comments and suggestions.

**Conflicts of Interest:** The authors declare that there is no conflict of interest regarding the publication of this paper.

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
