# Peer review of "Multi-Objective Quantum-Inspired Seagull Optimization Algorithm"

_electronics, doi:10.3390/electronics11121834_

Round 1

Reviewer 1 Report

This paper presented a multi-objective evolutionary algorithm combining quantum computing and seagull optimization algorithm to optimize the convergence and distribution of solutions in multi-objective optimization problems. The main contribution of the paper yields on the introduction of quantum computation features on population improving process. The proposed method is novel and has obtained superior performance in comparison to other competing algorithms. In general, the paper is well organized, and has certain innovation ability and reliable application scenarios. Some suggestions for improvement are as follows:

1.         The paper should be carefully rewritten in order to achieve enough level to be understandable.

2.         The contribution of this paper should be more clearly shown in the introduction section.

3.         Please explain and justify why you choose quantum-inspired method?

4.         The experimental results need to be expressed with more figures.

5.         What do μi and σi in Eq. (8) mean?

6.         Details about deciding whether to accept the current optimal solution as the global optimum need to be stated more clearly.

7.         Please express the role of specific correction in Eq. (23) more explicitly.

8.         What is the advantage of opposition-based learning in this algorithm?

Author Response

    Thank you for your letter and for the reviewer's comments concerning our manuscript entitled “Multi-Objective Quantum-Inspired Seagull Optimization Algorithm” (electronics-1763635). Those comments are all valuable and very helpful for revising and improving our paper, as well as the importance guiding significance to our researches.

    We have substantially revised our manuscript after reading the comments provided by the editor and reviewers, and revised portion are marked in red in the paper. 

Reviewer 2 Report

This paper proposes a multi-objective quantum-inspired seagull optimization algorithm (MOQSOA) for multi-objective optimization problems. The proposed algorithm is novel and has obtained superior performance in comparison to other competing algorithms. I recommend that this paper can be published after a major revision. My comments are as follows:

1. The writing needs further improvement.

2. The research motivation of this paper should be more clearly shown in introduction section.

3. It is suggested to add more figures to show the results of the experiments more intuitively.

4. What does the wave function in Eq. (8) do in the updation?

5. It is not necessary to present the performance metrics in the Section 4.1 since IGD and SP are widely known performance metrics for multi-objective optimization problems.

6. ZDT and DTLZ are not required to be described in Section 4.2, just cite relevant literatures.

7. Sections 4.1 and Section 4.2 can be merged.

8. The results of Wilcoxon signed-rank test are recommended to be placed in the table.

Author Response

(The authors gave the same response as above.)

Round 2

Reviewer 1 Report

All the comments have been addressed correctly and the paper is ready for publication in the present form.

Reviewer 2 Report

The authors have addressed all of my comments and the paper can be accepted in the present form.